# Endothelial cell-type-specific molecular requirements for angiogenesis drive fenestrated vessel development in the brain

Sweta Parab[1,2], Rachael E Quick[1,2], Ryota L Matsuoka[1,2]*

[1]Department of Cardiovascular and Metabolic Sciences, Lerner Research Institute, Cleveland Clinic, Cleveland, United States; [2]Cleveland Clinic Lerner College of Medicine of Case Western Reserve University, Cleveland, United States

**Abstract** Vascular endothelial cells (vECs) in the brain exhibit structural and functional heterogeneity. Fenestrated, permeable brain vasculature mediates neuroendocrine function, body-fluid regulation, and neural immune responses; however, its vascular formation remains poorly understood. Here, we show that specific combinations of vascular endothelial growth factors (Vegfs) are required to selectively drive fenestrated vessel formation in the zebrafish myelencephalic choroid plexus (mCP). We found that the combined, but not individual, loss of Vegfab, Vegfc, and Vegfd causes severely impaired mCP vascularization with little effect on neighboring non-fenestrated brain vessel formation, demonstrating fenestrated-vEC-specific angiogenic requirements. This Vegfs-mediated vessel-selective patterning also involves Ccbe1. Expression analyses, cell-type-specific ablation, and paracrine activity-deficient *vegfc* mutant characterization suggest that vEC-autonomous Vegfc and meningeal fibroblast-derived Vegfab and Vegfd are critical for mCP vascularization. These results define molecular cues and cell types critical for directing fenestrated CP vascularization and indicate that vECs' distinct molecular requirements for angiogenesis underlie brain vessel heterogeneity.

*For correspondence:
matsuor@ccf.org

Competing interests: The authors declare that no competing interests exist.

## Introduction

Cerebrovascular diseases remain a leading cause of long-term disability worldwide (*Tong et al., 2019*). Many of these diseases originate, at least in part, from genetic defects that impact the development and maturation of blood vessels in the brain and meninges. Understanding the molecular basis of brain and meningeal vascularization during development will help identify therapeutic targets for preventing or treating vascular abnormalities associated with these diseases. Current evidence shows that vascular endothelial cells (vECs) in the central nervous system (CNS) display heterogeneous structural and functional properties (*Augustin and Koh, 2017*; *Mastorakos and McGavern, 2019*; *O'Brown et al., 2018*). Two striking examples of this heterogeneity are the vECs that form the semipermeable blood-brain barrier (BBB) and those that develop highly permeable fenestrae. Despite substantial progress in our understanding of CNS angiogenesis and BBB formation/maintenance (*Andreone et al., 2017*; *Ben-Zvi et al., 2014*; *Daneman et al., 2009*; *Haigh et al., 2003*; *Junge et al., 2009*; *Kuhnert et al., 2010*; *Liebner et al., 2008*; *Raab et al., 2004*; *Segarra et al., 2018*; *Stenman et al., 2008*; *Vanhollebeke et al., 2015*; *Wang et al., 2018*; *Ye et al., 2009*), little is known about the molecular mechanisms directing fenestrated vessel formation in the choroid plexus (CP) and circumventricular organs. Identification of molecular cues that govern the vascularization of these organs will accelerate our understanding of cerebrovascular development and heterogeneity, providing critical insights into their pathological processes.

The CPs consist of epithelial tissue masses located within the brain ventricles and are vascularized with fenestrated capillaries that lack the BBB (*Bill and Korzh, 2014*; *Cserr, 1971*; *Damkier et al., 2013*; *Liddelow, 2015*). The CPs serve to produce cerebrospinal fluid (CSF) and remove metabolic wastes from the CNS (*Bill and Korzh, 2014*; *Cserr, 1971*; *Damkier et al., 2013*; *Liddelow, 2015*), thereby regulating the CNS microenvironment. They also serve as an important source of the biologically active molecules involved in brain development and repair (*Ghersi-Egea et al., 2018*), form the blood-CSF barrier (*Bill and Korzh, 2014*; *Cserr, 1971*; *Damkier et al., 2013*; *Liddelow, 2015*), provide immune surveillance, and regulate immune cell trafficking into the CSF in response to neuro-inflammation and brain injury/disease (*Lopes Pinheiro et al., 2015*; *Wilson et al., 2010*). These CP's crucial roles in brain development, homeostasis, and immunity are thought to require the establishment of fenestrated vasculature (*Ghersi-Egea et al., 2018*; *Liddelow, 2015*; *Lopes Pinheiro et al., 2015*). Although the morphogen Sonic hedgehog is indicated to be involved in vascular outgrowth during CP development (*Nielsen and Dymecki, 2010*), no molecular cue that drives fenestrated CP vascular development has been reported in any vertebrate species.

In mammals, CPs are located in the lateral, 3rd, and 4th ventricular brain surfaces (*Liddelow, 2015*). This limits tissue accessibility for 3D imaging, making it difficult to analyze the blood vessel(s) that gives rise to the CP vasculature in these animal models. Here, we employed the zebrafish model to overcome this technical limitation and investigate the molecular genetic mechanisms governing fenestrated CP vascular development.

## Results

### Zebrafish CPs and molecularly heterogeneous networks of the brain and meningeal vasculature

In zebrafish, the CPs form in two distinct anatomical locations close to the dorsal surfaces of the brain meninges and are termed as the diencephalic and myelencephalic CP (dCP and mCP) (*Figure 1A*). The mCP is larger in size and thought to be equivalent to the 4th ventricle CP in mammals due to its anatomical location being in close proximity to the 4th ventricle and cerebellum in zebrafish (*Bill and Korzh, 2014*; *García-Lecea et al., 2008*; *Henson et al., 2014*). As previously reported (*Henson et al., 2014*), we validated that the *Et(cp:EGFP)* enhancer trap line marks only epithelial cells in the dCP and mCP at 5 days post fertilization (dpf) (*Figure 1B*) since the EGFP⁺ cells in the mCP were outlined by Claudin-5 tight junction protein expression, a marker for CP epithelial cells (*Figure 1C*). Moreover, mCP vasculature in zebrafish exhibits the following molecular signatures of fenestrated vessels (*Umans et al., 2017*; *van Leeuwen et al., 2018*): (1) high expression of the structural protein PLVAP, an endothelial marker for the high permeability state and (2) low expression of GLUT1 and Claudin5, endothelial markers for the BBB state (*Figure 1D* and *Figure 1—figure supplement 1A*). These observations suggest that the molecular signatures of CP epithelial and endothelial cells are well conserved between zebrafish and mammals. Importantly, our transmission electron microscopy analyses further revealed that CP epithelial cells of 10 dpf zebrafish larvae displayed anatomical features of their mammalian counterparts at the ultrastructural level, including tight junctions, microvilli, cilia, and desmosomes (*Figure 1E–H'*), providing additional evidences to support the notion of interspecies conservation and thus making the zebrafish model suitable for molecular genetics studies.

Close examination of 6 and 10 dpf *Tg(plvap:EGFP);Tg(glut1b:mCherry)* double transgenic (Tg) reporter fish allowed us to observe a molecularly heterogeneous network of brain and meningeal vasculature; (1) blood vessels exhibiting high levels of *Tg(plvap:*EGFP), but low levels of *Tg(glut1b:*mCherry), expression; (2) those exhibiting low levels of *Tg(plvap:*EGFP), but high levels of *Tg(glut1b:*mCherry), expression; and (3) those with medium levels of both *Tg(plvap:*EGFP) and *Tg(glut1b:*mCherry) expression (*Figure 1D* and *Figure 1—figure supplement 1A*). In addition to their heterogeneous expression at the transcriptional level, we also observed the heterogeneous Claudin-5 protein expression in these brain and meningeal vessels (*Figure 1—figure supplement 1B–D''*). Intriguingly, these networks of brain vasculature are all interconnected, raising the question of how molecularly heterogeneous networks of brain vasculature arise during development.

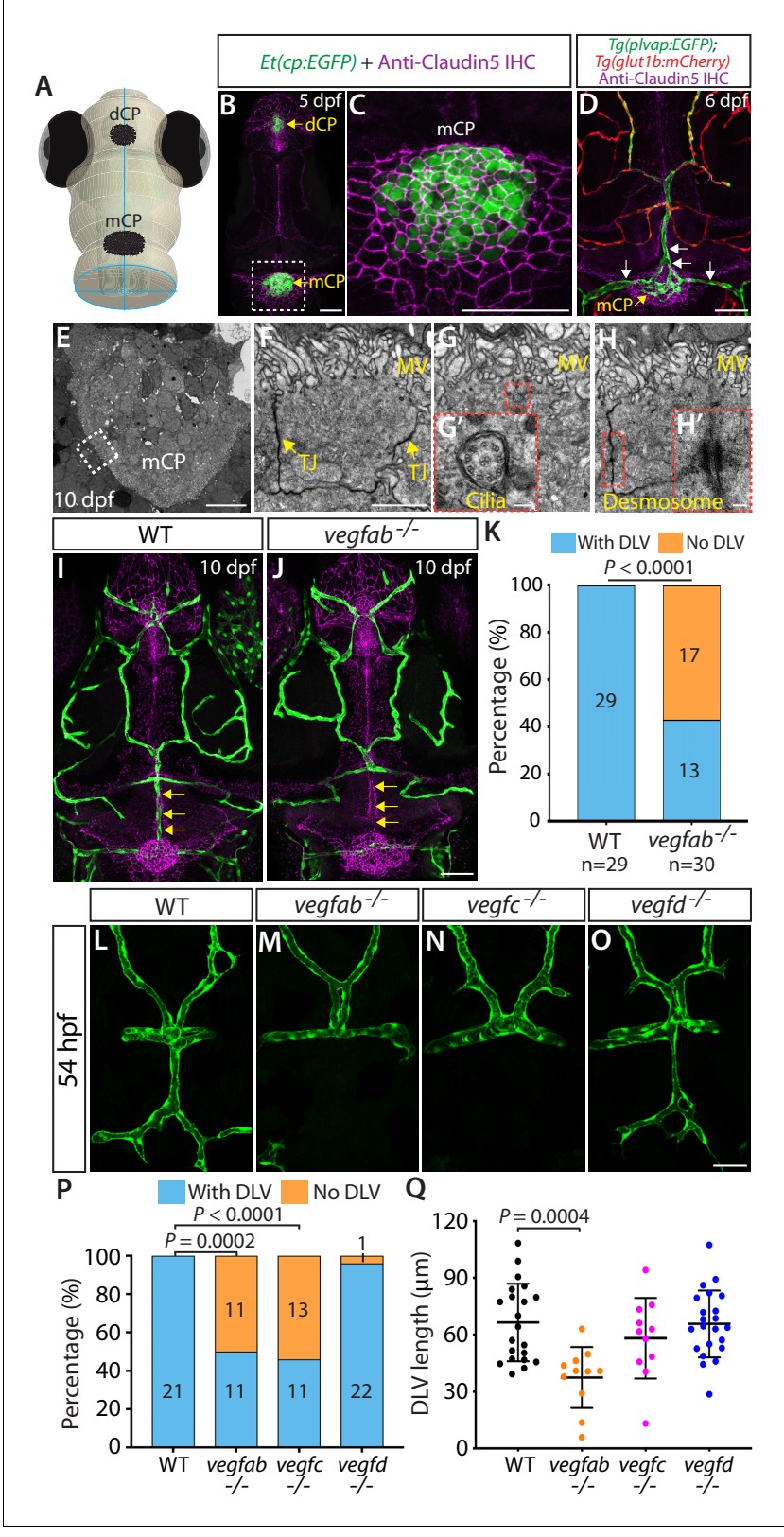

**Figure 1.** Fenestrated mCP vascular formation defects in *vegfab* and *vegfc* mutant zebrafish. (**A**) Schematic representation of the dorsal view of the zebrafish head, indicating the locations of the diencephalic and myelencephalic CP (dCP and mCP, respectively). (**B and C**) Dorsal views of a 5 dpf *Et(cp:EGFP)* head immunostained for Claudin-5 (magenta), indicating EGFP⁺ and Claudin-5⁺ mCP epithelial cells. A magnified image

*Figure 1 continued on next page*

*Figure 1 continued*

of the boxed area is shown in (C). (D) Dorsal view of a 6 dpf *Tg(plvap:EGFP);Tg(glut1b:mCherry)* head immunostained for Claudin-5 (magenta) shows heterogeneous *Tg(plvap*:EGFP) and *Tg(glut1b*:mCherry) expression patterns in the brain and meningeal vasculature. Strong *Tg(plvap*:EGFP) and weak *Tg(glut1b*:mCherry) expression was observed in the mCP vasculature (white arrows). (E–H) Transmission electron microscopy images of 10 dpf wild-type (WT) mCP and its surrounding tissues (E) and of mCP epithelial cells (F–H). Magnified images of mCP epithelial cells (F–H) were taken in areas between the mCP and its neighboring tissues as indicated by the dashed line. mCP epithelial cells showed tight junctions (TJ) and microvilli (MV), the 9 + 2 arrangement of microtubules of cilia (G'), and desmosomes (H'). Magnified images of the boxed areas (G and H) are shown in (G') and (H'), respectively. (I and J) Dorsal views of 10 dpf WT (I) and *vegfab*-/- (J) cranial vasculature visualized by *Tg(kdrl*:EGFP) expression and anti-Claudin-5 immunostaining (magenta) show a specific loss of the dorsal longitudinal vein (DLV) in *vegfab*-/- larvae (yellow arrows). (K) Percentage of the fish of indicated genotype with and without the DLV at 10 dpf (n = 29 for WT and n = 30 for *vegfab*-/- fish). (L–O) Dorsal views of 54 hpf WT (L), *vegfab*-/- (M), *vegfc*-/- (N), and *vegfd*-/- (O) cranial vasculature visualized by *Tg(kdrl*:EGFP) expression. Approximately half of the *vegfab*-/- and *vegfc*-/- embryos examined lacked the DLV. (P) Percentage of the fish of indicated genotype with and without the DLV at 54 hpf (n = 21 for WT, n = 22 for *vegfab*-/-, n = 24 for *vegfc*-/-, and n = 23 for *vegfd*-/- fish). (Q) Quantification of DLV lengths of the fish that formed the DLV at 54 hpf (n = 21 for WT, n = 11 for *vegfab*-/-, n = 11 for *vegfc*-/-, and n = 22 for *vegfd*-/- fish). Data are means ± SD. Scale bars: 50 μm in (B), (C), (D), (J), (O); 10 μm in (E); 2 μm in (F); 100 nm in (G') and (H').

The online version of this article includes the following source data and figure supplement(s) for figure 1:

**Source data 1.** Quantifications of DLV lengths and the 'No DLV' phenotype in 54 hpf WT and *vegf* mutant embryos.
**Figure supplement 1.** Molecularly heterogeneous networks of the brain and meningeal vasculature and developmental time courses of mCP vascularization.
**Figure supplement 2.** Characterization of DLV formation defects in *vegfab* and *vegfc* mutants at distinct larval stages and visualization of perfused cranial vasculature in WT and *vegfab* mutant larvae.
**Figure supplement 3.** Heatshock-induced overexpression of sFlt1 and sFlt4 partially phenocopied the DLV formation defects observed in *vegfab* and *vegfc* mutants, respectively.

## Identification of angiogenic cues required for mCP vascular development

In order to explore molecular cues that direct CP vascularization, we screened the loss-of-function mutant zebrafish lacking Vegf activity because VEGF signaling has been indicated to be crucial for the maintenance of CP vascular perfusion and fenestrae in adult mice (*Kamba et al., 2006*; *Maharaj et al., 2008*). Since *vegfaa*bns1 mutants died at approximately 5 dpf due to severe early vascular formation defects (*Rossi et al., 2016*), we focused our analysis on the *vegfab*bns92, *vegfc*hu6410, and *vegfd*bns257 mutants carrying a *Tg(kdrl:EGFP)* endothelial cell-specific reporter at 10 dpf – a developmental stage when most of the major blood vessels are formed (*Isogai et al., 2001*). Intriguingly, we found that nearly 57% of 10 dpf *vegfab* mutants lacked the dorsal longitudinal vein (DLV), a main blood vessel that supplies blood to the mCP (*Bill and Korzh, 2014*; *Figure 1J and K*), while all of the wild-type (WT) larvae, as well as a vast majority of the *vegfc* and *vegfd* mutants, examined at 10 dpf formed this vessel (*Figure 1I and K*; data not shown). Notably, while not fully penetrant in 10 dpf *vegfab* mutants, this vascular phenotype is prominent and specific to this vessel since most other brain and meningeal vessels formed (*Figure 1I and J*).

To determine whether this DLV defect in *vegfab* mutants is a developmental or maintenance deficit, we conducted their phenotypic analyses at various developmental stages. As previously indicated (*Bill and Korzh, 2014*), we observed that (1) initial DLV sprouting occurs around 45 hours post fertilization (hpf); (2) the sprouting DLV extends toward the mCP and branches to begin joining the bilateral posterior cerebral vein (PCeV) around 54 hpf; and (3) the DLV and bilateral PCeV converge to form the mCP vasculature, which is nearly complete by 72 hpf (*Figure 1—figure supplement 1E–I*). To assess DLV formation in mutants, we used two different measurements. The first quantified the percentage of fish that lacked the DLV. The second measured the length of the DLV in all remaining fish that formed the DLV. At 54 hpf, while all the WT embryos formed the DLV (*Figure 1L*), we found that approximately 50% of *vegfab* and *vegfc* mutants lacked the DLV (*Figure 1M,N,P*). Most *vegfd* mutants formed the DLV (*Figure 1O,P*), and only 4% showed the 'No DLV' phenotype. We also found that DLV lengths were significantly shorter in *vegfab* mutants than in WT (*Figure 1Q*). While *vegfc* mutants displayed a lower percentage of the 'No DLV' phenotype as development proceeded

at 102 and 154 hpf (41% and 12%), we observed that nearly 50–60% of *vegfab* mutants continue to exhibit this phenotype (*Figure 1—figure supplement 2A–F*). Microangiography analysis of the circulatory system by injecting Qdot 655 streptavidin conjugated nanocrystals into the common cardinal vein of *vegfab* mutants revealed the absence of a perfused, functional DLV in the mutants that lack the *Tg(kdrl:EGFP)* endothelial reporter expression in this vessel (*Figure 1—figure supplement 2G–H''*). Moreover, under transmitted differential interference contrast imaging, these *vegfab* mutants displayed no blood flow or circulating blood cells in the region where the DLV normally forms (data not shown). These results showed that the 'No DLV' phenotype is a developmental defect leading to the complete absence of a perfused DLV and that Vegfab and Vegfc are crucial for DLV formation.

Consistent with these mutant results, we found that the heat-shock induced overexpression of a soluble form of Flt1 and Flt4 (sFlt1 and sFlt4), decoy receptors for Vegfa and Vegfc, respectively, partially recapitulated the DLV formation defects observed in *vegfab* and *vegfc* mutants (*Figure 1—figure supplement 3*). In this experiment, we utilized the *Tg(hsp70l:sflt1)* and *Tg(hsp70l:sflt4)* lines which, upon a heat-shock treatment, over-express sFlt1 and sFlt4 respectively, allowing for the inhibition of Vegfa- or Vegfc-induced signaling within limited timeframes (*Matsuoka et al., 2016*; *Matsuoka et al., 2017*). To ensure that overexpression of sFlt1 or sFlt4 begins prior to the onset of DLV sprouting and is maintained afterwards until imaging analysis, we gave fish multiple heat-shocks every 8 hr starting at 41 hpf. We found that while the heat-shocked *Tg(hsp70l:sflt1)* larvae did not display significantly higher penetrance of the 'No DLV' phenotype (21%) than that of non-heat-shocked (0%) and heat-shocked (10%) control fish (*Figure 1—figure supplement 3H*), DLV lengths were significantly shorter in the rest of the *Tg(hsp70l:sflt1)* larvae that formed the DLV (*Figure 1—figure supplement 3J*), partially phenocopying *vegfab* mutant DLV phenotypes (*Figure 1P,Q*). In addition, we also observed a significantly higher percentage of the 'No DLV' phenotype (33%) in *Tg(hsp70l:sflt4)* larvae after the same heat-shock treatments (*Figure 1—figure supplement 3I*), although the rest of the *Tg(hsp70l:sflt4)* larvae formed a DLV of comparable lengths to those observed in control animals (*Figure 1—figure supplement 3K*), which is consistent with *vegfc* mutant DLV phenotypes (*Figure 1P,Q*). These results suggest that temporally restricted inhibition of Vegfab- or Vegfc-dependent signaling during DLV formation is sufficient to abrogate this process.

## Multiple Vegf ligands function redundantly to drive mCP vascularization

The partially penetrant 'No DLV' phenotype observed in *vegfab* and *vegfc* mutants led us to hypothesize that Vegfab and Vegfc are functionally redundant in regulating this vascular process. To test this hypothesis, we set up incrosses of *vegfab;vegfc* double heterozygous adults and analyzed DLV formation in all progeny generated from these crosses (*Figure 2A*). We chose to analyze larvae at 96 hpf to minimize the effects of potential developmental delays in *vegfab;vegfc* double mutants. These analyses revealed that *vegfab;vegfc* double homozygous mutants displayed markedly enhanced penetrance and expressivity of mCP vascularization defects. Nearly 81% of *vegfab^-/-^;vegfc^-/-^* larvae exhibited the 'No DLV' phenotype at 96 hpf (*Figure 2D–F*). This penetrance is significantly higher than their *vegfab^-/-^* (38%) and *vegfc^-/-^* (16%) siblings (*Figure 2F*). Moreover, we found that *vegfab^+/-^; vegfc^+/-^* larvae exhibited approximately threefold higher penetrance of this phenotype (22%) than their sibling *vegfab^+/-^* (7%) and *vegfc^+/-^* (8%) fish (*Figure 2F*). These results showed that *vegfab* and *vegfc* genetically interact in DLV formation.

It is possible that impaired DLV formation is caused by the defective formation of the neighboring blood vessels that give rise to the DLV. To address this possibility, we analyzed three of the DLV's neighboring blood vessels (*Figure 2B*), including the PCeV, middle cerebral veins (MCeV), and mesencephalic veins (MsV). Importantly, the MCeV and MsV exhibit molecular signatures shared by non-fenestrated vasculature (*Figures 1D* and *2C*, *Figure 1—figure supplement 1A–C''*), while the PCeV displays those of fenestrated vessels (*Figures 1D* and *2C'* and *Figure 1—figure supplement 1A–D''*; *Umans et al., 2017*; *van Leeuwen et al., 2018*). Intriguingly, we found that PCeV formation defects were markedly enhanced in *vegfab;vegfc* double mutants when compared to their respective single mutants, while this was not the case with the formation of the MCeV and MsV (*Figure 2G*). These results indicate that the redundant actions of Vegfab/Vegfc differentially regulate the formation of the brain blood vessels that exhibit distinct molecular signatures.

To investigate whether Vegfab/Vegfc-dependent angiogenesis is central to DLV and PCeV formation, we examined *vegfab;vegfd* double homozygous mutants at 96 hpf for comparison (*Figure 2—*

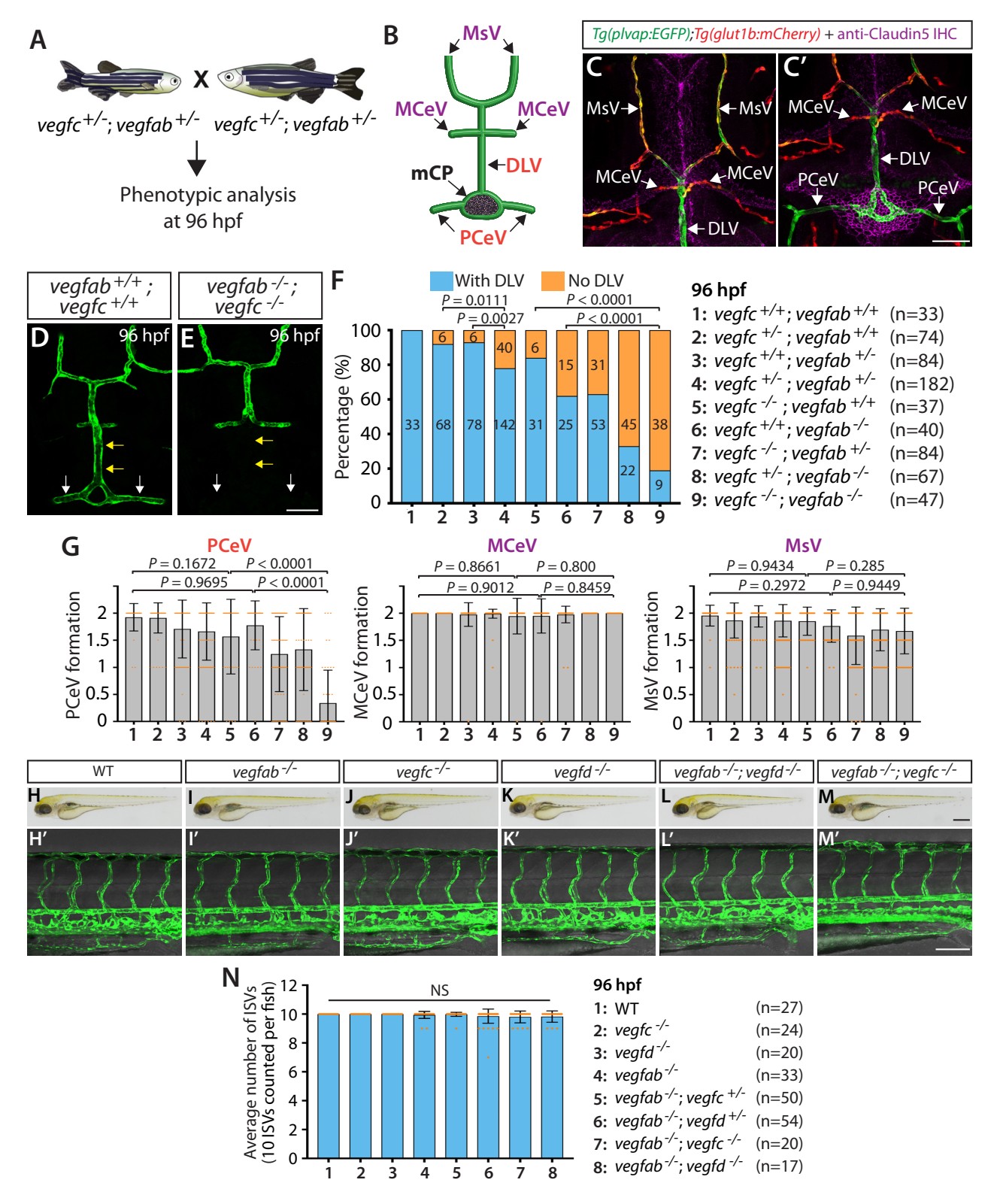

**Figure 2.** *vegfab* genetically interacts with *vegfc* in mCP vascular development. (**A**) Experimental setup for the results shown in (**D–G**). (**B**) Schematic representation of the dorsal view of 96 hpf WT cranial vasculature, illustrating distinct cranial vessels. PCeV: posterior cerebral vein, MCeV: middle cerebral vein, MsV: mesencephalic vein. (**C and C'**) Dorsal views of a 10 dpf *Tg(plvap:EGFP);Tg(glut1b:mCherry)* head immunostained for Claudin-5 (magenta) show that the DLV and PCeV exhibit stronger *Tg(plvap:*EGFP), and weaker *Tg(glut1b:*mCherry), expression than the MCeV and MsV do. (D

*Figure 2 continued on next page*

Figure 2 continued

and E) Dorsal views of 96 hpf *vegfab*$^{+/+}$;*vegfc*$^{+/+}$ (D) and *vegfab*$^{-/-}$;*vegfc*$^{-/-}$ (E) cranial vasculature visualized by *Tg(kdrl*:EGFP) expression show a specific loss of the DLV (yellow arrows) and PCeV (white arrows) in *vegfab*$^{-/-}$;*vegfc*$^{-/-}$ larvae. (F) Percentage of 96 hpf fish of indicated genotype with and without the DLV (the number of the animals examined per genotype is listed in the panel). (G) Quantification of PCeV, MCeV, and MsV formation at 96 hpf. PCeV formation was severely and selectively compromised in *vegfc*$^{-/-}$;*vegfab*$^{-/-}$ larvae compared to the corresponding single mutants. (H–M) Brightfield images of 96 hpf WT (H), *vegfab*$^{-/-}$ (I), *vegfc*$^{-/-}$ (J), *vegfd*$^{-/-}$ (K), *vegfab*$^{-/-}$;*vegfd*$^{-/-}$ (L), and *vegfab*$^{-/-}$;*vegfc*$^{-/-}$ (M) larvae. (H'–M') Lateral views of 96 hpf WT (H'), *vegfab*$^{-/-}$ (I'), *vegfc*$^{-/-}$ (J'), *vegfd*$^{-/-}$ (K'), *vegfab*$^{-/-}$;*vegfd*$^{-/-}$ (L'), and *vegfab*$^{-/-}$;*vegfc*$^{-/-}$ (M') larval trunk vasculature visualized by *Tg(kdrl*:EGFP) expression. (N) Quantification of average number of ISVs in 96 hpf fish of indicated genotype (the number of the animals examined per genotype is listed in the panel). No significant difference in the number of ISVs was observed across the genotypes. NS: not significant. Data are means ± SD. Scale bars: 50 μm in (C'), (E), and (M'); 1 mm in (M).

The online version of this article includes the following figure supplement(s) for figure 2:

**Figure supplement 1.** *vegfab* genetically interacts with *vegfd* in mCP vascular development.

figure supplement 1). Surprisingly, despite the fact that *vegfd* mutants alone did not exhibit a significant increase in the 'No DLV' phenotype (12.5%) compared to the WT larvae of the same age (0%) (*Figure 2F* and *Figure 2—figure supplement 1B*), genetic deletion of one or both copies of *vegfd* in the *vegfab* mutant background dramatically increased the penetrance of this phenotype (88% for *vegfab*$^{-/-}$;*vegfd*$^{+/-}$ larvae; 97% for *vegfab*$^{-/-}$;*vegfd*$^{-/-}$ larvae) compared to that observed in their *vegfab*$^{-/-}$ siblings (69%) (*Figure 2—figure supplement 1B*). We found that *vegfab*$^{-/-}$;*vegfd*$^{-/-}$ larvae also exhibited a pronounced defect in PCeV formation at 96 hpf, while only a mild defect in MsV, but not MCeV, formation was observed at this age (*Figure 2—figure supplement 1C*). It is important to note that *vegfab*$^{-/-}$;*vegfd*$^{-/-}$ and *vegfab*$^{-/-}$;*vegfc*$^{-/-}$ larvae did not exhibit apparent differences in body size and/or gross morphology (*Figure 2H–M*).

## Genetic elimination of specific pairs of *vegf* genes impacts mCP vascularization

Remarkably, the severe DLV and PCeV formation defects in *vegfab*$^{-/-}$;*vegfc*$^{-/-}$ and *vegfab*$^{-/-}$;*vegfd*$^{-/-}$ larvae were observed even at 10 dpf (*Figure 3C,D*). The results showed a tendency toward slightly lower penetrance of the 'No DLV' phenotype across genotypes at 10 dpf (*Figure 3F*) as compared to those at 96 hpf (*Figure 2F* and *Figure 2—figure supplement 1B*). This tendency may indicate that some of the larvae can recover from the lack of the DLV between 4 and 10 dpf. Nonetheless, 10 dpf *vegfab*$^{-/-}$;*vegfc*$^{-/-}$ and *vegfab*$^{-/-}$;*vegfd*$^{-/-}$ larvae showed markedly higher penetrance of this phenotype (65% and 76%, respectively) relative to their *vegfab*$^{-/-}$ siblings (38%) (*Figure 3F*). Similarly, we observed severe defects in PCeV formation in *vegfab*$^{-/-}$;*vegfc*$^{-/-}$ and *vegfab*$^{-/-}$;*vegfd*$^{-/-}$ larvae, although none of the *vegfab*, *vegfc*, and *vegfd* single mutant fish examined at this stage showed this defect when compared to WT fish (*Figure 3G*). *vegfab*$^{-/-}$;*vegfc*$^{-/-}$ and *vegfab*$^{-/-}$;*vegfd*$^{-/-}$ larvae displayed mild defects in MsV formation when compared to WT fish; however, no significant differences in MCeV and MsV formation were observed in these double mutants compared to *vegfab*$^{-/-}$ fish (*Figure 3G*). These results demonstrate that *vegfab* genetically interacts with *vegfc* and *vegfd* specifically in DLV and PCeV formation around this area of the brain.

To test whether *vegfc* and *vegfd* genetically interact in mCP vascularization, we examined *vegfc*$^{-/-}$;*vegfd*$^{-/-}$ larvae at 10 dpf for comparison. These double mutants exhibited slightly increased 'No DLV' phenotype (16%) when compared to WT fish (0%), but did not show significantly enhanced penetrance when compared to *vegfd*$^{-/-}$ larvae (9%) (*Figure 3F*). Moreover, most of the *vegfc*$^{-/-}$;*vegfd*$^{-/-}$ larvae examined formed the mCP and neighboring vasculature and did not display a significant difference in PCeV formation when compared to WT fish (*Figure 3B,F,G*). These results demonstrate that the genetic elimination of only the specific pairs of *vegf* genes has a profound impact on mCP vascularization and indicate that Vegfc and Vegfd play modulatory roles in regulating this process.

Since *vegfab*$^{-/-}$;*vegfc*$^{-/-}$ and *vegfab*$^{-/-}$;*vegfd*$^{-/-}$ larvae did not display a fully penetrant 'No DLV' phenotype, we next analyzed *vegfab*;*vegfc*;*vegfd* triple mutants at 10 dpf. We found that all the triple mutants examined (n = 16) lacked the DLV as well as the PCeV, except for one larva (*Figure 3E–G*), demonstrating significantly enhanced penetrance of these phenotypes when compared to *vegfab*$^{-/-}$ fish. However, this was not the case with MCeV and MsV formation. These extensive genetic

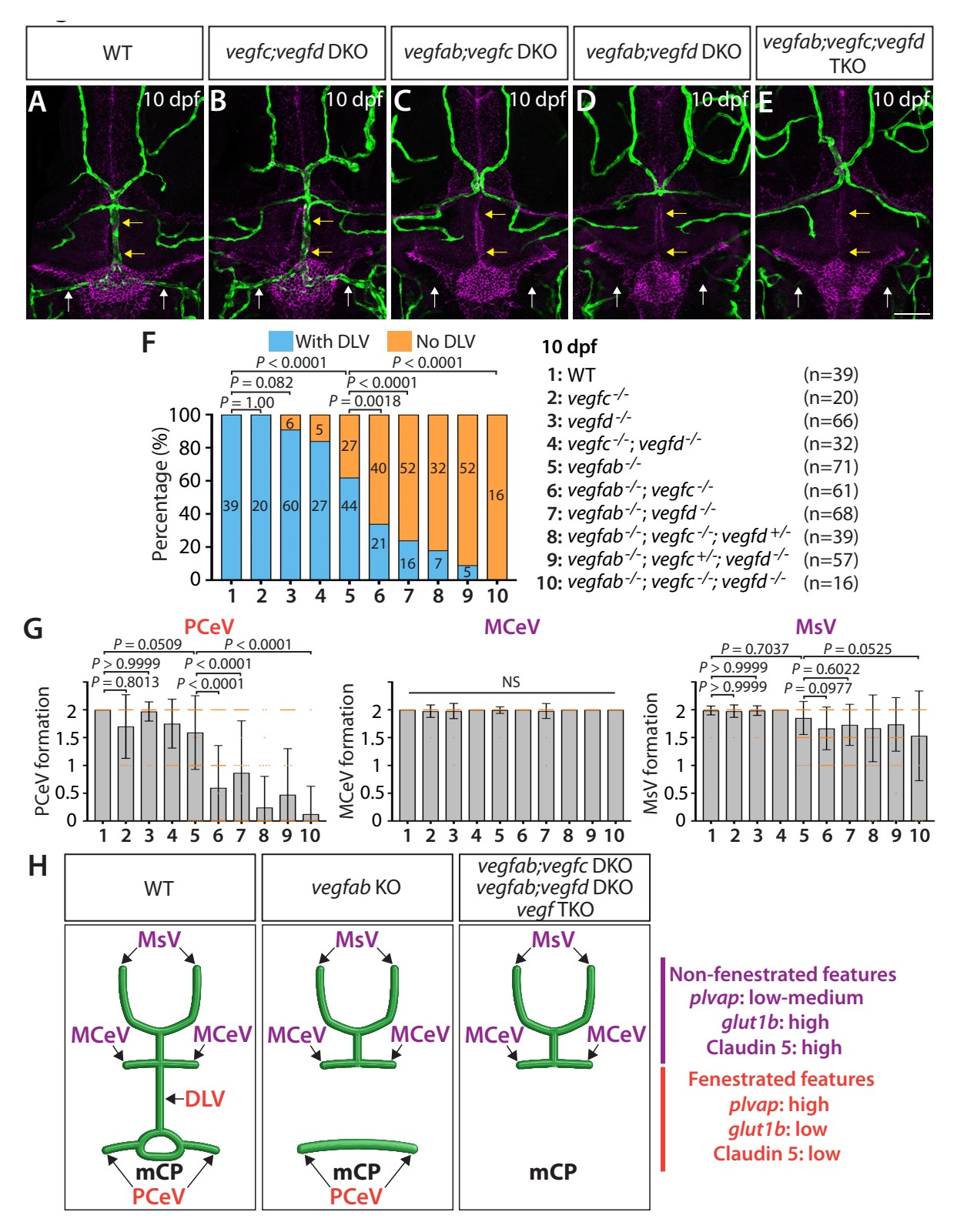

**Figure 3.** Multiple Vegf ligands function redundantly to drive mCP vascularization. (A–E) Dorsal views of 10 dpf WT (A), *vegfc⁻/⁻;vegfd⁻/⁻* (*vegfc;vegfd* DKO, B), *vegfab⁻/⁻;vegfc⁻/⁻* (*vegfab;vegfc* DKO, C), *vegfab⁻/⁻;vegfd⁻/⁻* (*vegfab;vegfd* DKO, D), and *vegfab⁻/⁻;vegfc⁻/⁻;vegfd⁻/⁻* (*vegfab;vegfc;vegfd* TKO, E) cranial vasculature visualized by *Tg(kdrl*:EGFP) expression and anti-Claudin-5 immunostaining (magenta) show specific losses of the DLV (yellow arrows) and PCeV (white arrows) in *vegfab⁻/⁻;vegfc⁻/⁻* (C), *vegfab⁻/⁻;vegfd⁻/⁻* (D), and *vegfab⁻/⁻;vegfc⁻/⁻;vegfd⁻/⁻* (E) larvae. (F) Percentage of 10 dpf fish of indicated

*Figure 3 continued on next page*

Figure 3 continued

genotype with and without the DLV (the number of the animals examined per genotype is listed in the panel). (G) Quantification of PCeV, MCeV, and MsV formation at 10 dpf. (H) Schematic representations of the dorsal view of 10 dpf WT, *vegfab*$^{-/-}$ (*vegfab* KO), *vegfab*$^{-/-}$;*vegfc*$^{-/-}$ (*vegfab*;*vegfc* DKO), *vegfab*$^{-/-}$;*vegfd*$^{-/-}$ (*vegfab*;*vegfd* DKO), and *vegfab*$^{-/-}$;*vegfc*$^{-/-}$;*vegfd*$^{-/-}$ (*vegf* TKO) cranial vasculature summarize the vascular phenotypes observed in these mutants. Genetic evidence shows that there are specific requirements for Vegfab/Vegfc/Vegfd-dependent angiogenesis involved in fenestrated mCP vascular development. Scale bar: 50 μm.

The online version of this article includes the following figure supplement(s) for figure 3:

**Figure supplement 1.** Developmental relationships between FGPs and mCP vasculature in WT, *vegfab*$^{-/-}$, and *vegfc*$^{-/-}$;*vegfd*$^{-/-}$ larvae.

analyses provide evidence that the functional redundancy among these three Vegf ligands selectively directs fenestrated DLV and PCeV formation leading to mCP vascularization (*Figure 3H*).

## Combined genetic inactivation of *ccbe1* and *vegfab* leads to significantly enhanced defects in DLV and PCeV formation

Prior studies have shown that Ccbe1, Adamts3, and Adamts14 serve to proteolytically cleave and convert the pro-active form of Vegfc into its mature form during lymphangiogenesis in zebrafish (*Hogan et al., 2009*; *Le Guen et al., 2014*; *Wang et al., 2020*). Among these three proteins, *ccbe1* appears to be highly expressed in the developing brain and meningeal compartments based on its BAC transgenic reporter expression at 48 hpf (*Wang et al., 2020*), while *adamts3* BAC transgenic reporter expression appear to be very low at the same stage. Since 48 hpf is a developmental time point when the DLV is extending toward the mCP, *ccbe1* could be a strong candidate that acts to activate Vegfc in this area of the brain.

To investigate a role for Ccbe1 in mCP vascularization, we took the approach of injecting ribonucleoprotein (RNP) complexes that are composed of *ccbe1*-specific CRISPR RNA (crRNA), trans-activating crRNA (tracrRNA), and Cas9 protein. This approach using crRNA:tracrRNA:Cas9 protein RNP complexes was previously shown to be a highly efficient method to achieve bi-allelic inactivation of target genes when injected at the one-cell stage, which allows for the generation of F0 embryos that recapitulate homozygous mutant phenotypes (*Hoshijima et al., 2019*). We tested this method and observed that simultaneous injection of three different crRNA RNP complexes targeted to a single gene works in a much more efficient manner to generate F0 embryos that lack target gene function and recapitulate homozygous mutant phenotypes (data not shown). Since it has been indicated that Ccbe1's full lymphangiogenic activity in vivo requires its EGF and collagen domains (*Roukens et al., 2015*), we designed three different crRNAs, all of which targeted the exons that encode the N-terminal domain of the Ccbe1 protein-coding sequences prior to the EGF domain (*Figure 4A*). Each of the designed crRNAs is thus expected to disrupt the function of Ccbe1 that is exerted through its EGF and collagen domains.

We first validated the efficacy of this triple crRNA RNP complex injection by examining the formation of fluorescent granular perithelial cells (FGPs) – also called Mato cells (*Venero Galanternik et al., 2017*), meningeal mural Lymphatic Endothelial Cells (muLECs) (*Bower et al., 2017*), or Brain Lymphatic Endothelial cells (BLECs) (*van Lessen et al., 2017*) – in the dorsal surfaces of the brain meninges (*Figure 4B–D*). We refer to this cell type as FGPs hereafter. We chose to analyze FGP formation because *ccbe1* mutants exhibit a severe defect in this process as shown in *vegfc* null mutants (*van Lessen et al., 2017*). To visualize FGPs, we used a *Tg(lyve1:DsRed)* reporter transgenic line. We observed that while all the uninjected and control Cas9 protein-injected *Tg(kdrl:EGFP);Tg(lyve1:DsRed)* larvae formed a bilateral loop of *Tg(lyve1:*DsRed)$^+$ FGPs in the dorsal surfaces of the brain meninges at 120 hpf, all the triple crRNA-injected larvae we examined completely lacked these FGP cells (*Figure 4C,D*). These control and crRNA-injected larvae did not display apparent differences in their gross morphology (*Figure 4C*, brightfield panels). Given the fact that a vast majority of 5 dpf *ccbe1* heterozygous larvae form meningeal FGPs similar to WT fish (*van Lessen et al., 2017*), we validated that this approach efficiently achieves bi-allelic genetic inactivation of *ccbe1*.

We hypothesized that if Vegfc requires Ccbe1-mediated proteolytic activation, these triple crRNA injected embryos will lack Vegfc activation, thus recapitulating *vegfc* null mutant phenotypes. To test this hypothesis, we first analyzed DLV formation in the triple crRNA injected embryos at 54 hpf (*Figure 4E*), a developmental time point at which nearly 54% of *vegfc* mutants exhibited the 'No

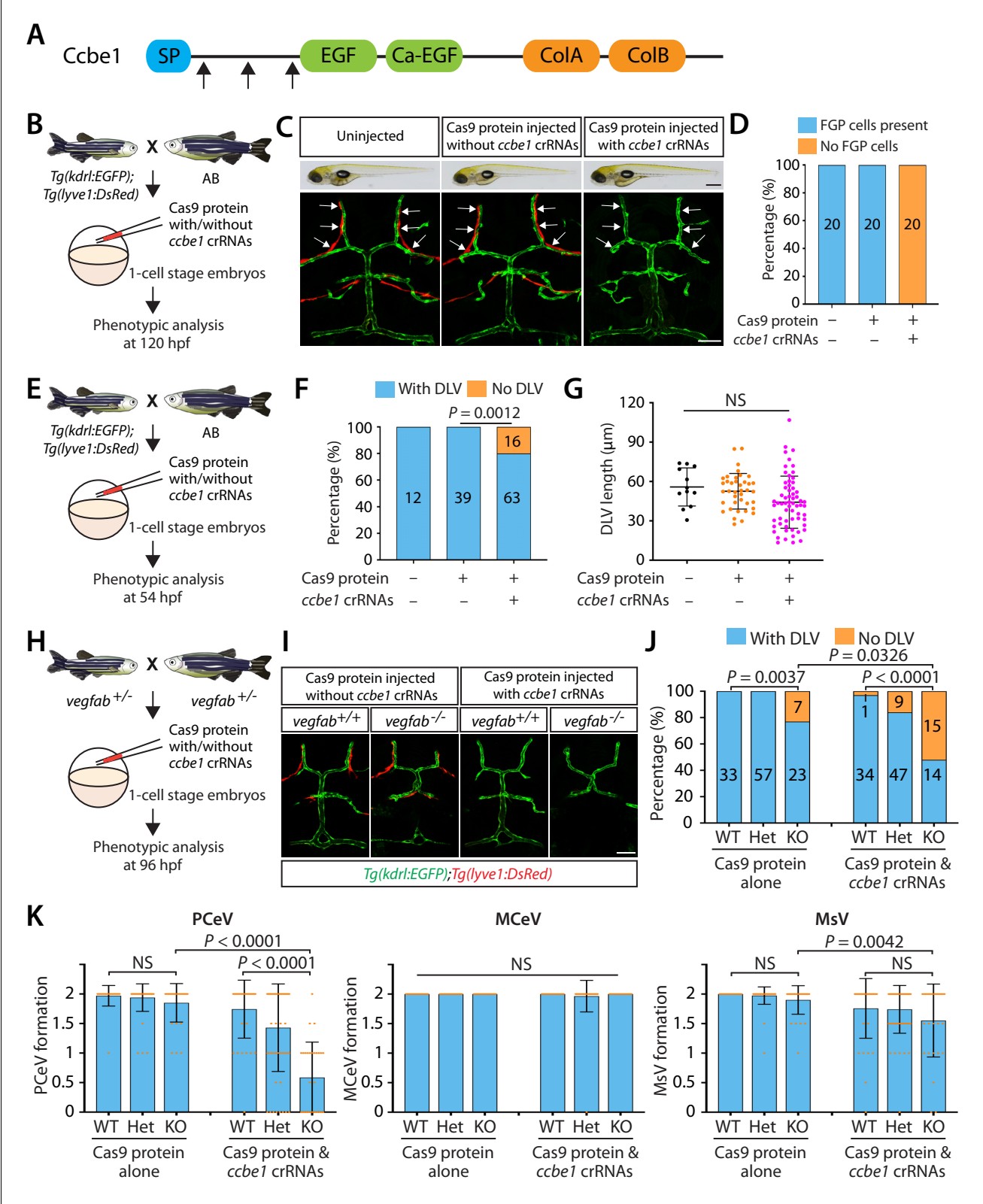

**Figure 4.** Combined genetic inactivation of *ccbe1* and *vegfab* leads to significantly enhanced defects in DLV and PCeV formation. (**A**) Predicted domain structure of zebrafish Ccbe1. Ccbe1 consists of a signal peptide (SP), an EGF domain, a calcium-binding EGF domain (Ca-EGF), and two collagen repeat domains (ColA and ColB). Arrows indicate the approximate positions of the protein sequences corresponding to the target sequences of the three designed CRISPR RNA (crRNA). (**B**) Experimental flows of the microinjection experiments for panels (**C and D**). Injection cocktails containing

*Figure 4 continued on next page*

Figure 4 continued

Cas9 protein with or without the three *ccbe1* crRNA:tracrRNA duplex complexes were injected into one-cell stage embryos produced from *Tg(kdrl: EGFP);Tg(lyve1:DsRed)* fish crossed with wild-type AB counterparts. Injected progeny were analyzed at 120 hpf for the presence or absence of *Tg(lyve1: DsRed)*[+] FGPs in the dorsal meningeal surfaces of the optic tectum. (C) Brightfield and confocal images of 120 hpf *Tg(kdrl:EGFP);Tg(lyve1:DsRed)* larvae after no injection or Cas9 protein injection with and without the three *ccbe1* crRNA:tracrRNA complexes. Although the larvae injected with the *ccbe1* crRNA ribonucleoprotein (RNP) complexes exhibited no apparent differences in their gross morphology compared to uninjected and Cas9 injected controls, they completely lacked *Tg(lyve1:DsRed)*[+] FGPs (white arrows) in the dorsal meningeal surfaces of the optic tectum. (D) Percentage of 120 hpf larvae of indicated treatment with and without *Tg(lyve1:DsRed)*[+] FGPs in the dorsal meningeal surfaces of the optic tectum (n = 20 animals examined for each group). (E) Experimental flows of the microinjection experiments for panel (F and G). (F) Percentage of 54 hpf larvae of indicated treatment with and without the DLV (the number of the animals examined per treatment is listed in the panel). A significant fraction of the embryos injected with the *ccbe1* crRNA RNP complexes lacked the DLV at this stage. (G) Quantification of DLV lengths of the fish of indicated treatment that formed the DLV at 54 hpf. (H) Experimental flows of the microinjection experiments for panels (I–K). (I) Dorsal views of 96 hpf *vegfab*[+/+] and *vegfab*[-/-] larvae that carried *Tg(kdrl:EGFP)* and *Tg(lyve1:DsRed)* transgenes after Cas9 protein injection with and without the three *ccbe1* crRNA:tracrRNA complexes. While control solution injected *vegfab*[+/+] and *vegfab*[-/-] larvae formed *Tg(lyve1:DsRed)*[+] FGPs in the dorsal surfaces of the brain meninges, those injected with the *ccbe1* crRNA RNP complexes completely lacked FGPs regardless of *vegfab* genotypes. A significant fraction of *vegfab*[-/-] larvae in both injection groups exhibited the 'No DLV' phenotype. (J) Percentage of 96 hpf larvae of indicated genotype and treatment with and without the DLV (the number of the animals examined per genotype is listed in the panel). *vegfab*[-/-] larvae injected with the *ccbe1* crRNA RNP complexes displayed a markedly increased penetrance of the 'No DLV' phenotype. (K) Quantification of PCeV, MCeV, and MsV formation at 96 hpf. *vegfab*[-/-] larvae injected with the Cas9 control solution as well as *vegfab*[+/+] larvae injected with the *ccbe1* crRNA RNP complexes did not exhibit a defect in PCeV formation. However, *vegfab*[-/-] larvae injected with the *ccbe1* crRNA RNP complexes exhibited severe defects in PCeV formation. Scale bars: 50 μm in (C) (fluorescence image), (J); 1 mm in (C) (brightfield image).

The online version of this article includes the following source data for figure 4:

**Source data 1.** Quantifications of DLV lengths and the 'No DLV' phenotype in 54 hpf *ccbe1* RNP-injected embryos and their sibling controls.

DLV' phenotype (*Figure 1P*). We found that approximately 20% of the triple crRNA-injected embryos displayed the 'No DLV' phenotype, while uninjected and control Cas9 protein-injected siblings all formed the DLV (*Figure 4F*). Similar to *vegfc* mutants examined at this developmental stage, the rest of the crRNA-injected embryos that formed the DLV did not exhibit DLV lengths shorter than those of the controls (*Figure 4G*).

Since *vegfab* and *vegfc* genetically interact in mCP vascular development, we next asked whether *vegfab* and *ccbe1* also genetically interact in this process. To achieve combined genetic inactivation of *vegfab* and *ccbe1*, we injected the triple crRNA RNPs into one-cell stage embryos generated from *vegfab*[+/-] incrosses (*Figure 4H*). We injected a solution containing Cas9 protein without the crRNA into their sibling embryos for comparison. We analyzed these injected larvae at 96 hpf to compare with the data of the larvae generated from *vegfab;vegfc* double heterozygous fish incrosses (*Figure 2A–G*). We simultaneously visualized *Tg(kdrl:EGFP)* and *Tg(lyve1:DsRed)* expression to analyze the formation of blood vessels and FGPs in every injected embryo. At 96 hpf, *Tg (lyve1:DsRed)*[+] FGPs were present in the control solution injected larvae regardless of their *vegfab* genotypes, while those injected with the *ccbe1* RNPs completely lacked these cells (*Figure 4I*), showing that the *ccbe1* RNPs-injected larvae subjected to this analysis also received effective gene inactivation of *ccbe1*. Intriguingly, similar to what we observed in *vegfab*[-/-];*vegfc*[-/-] larvae, we found that *vegfab*[-/-] larvae injected with the *ccbe1* RNPs exhibit a markedly increased penetrance of the 'No DLV' phenotype (52%) as compared to their *vegfab*[-/-] siblings injected with the control solution (23%) or the *ccbe1* RNPs-injected *vegfab*[+/+] siblings (3%) (*Figure 4J*). Strikingly enhanced defects were also observed in PCeV formation in the *ccbe1* RNPs-injected *vegfab*[-/-] larvae, while this was not the case for MCeV and MsV formation (*Figure 4K*). These results show genetic interactions between *vegfab* and *ccbe1* in DLV and PCeV formation, identifying Ccbe1 as an important molecular player involved in the Vegfab/Vegfc/Vegfd-dependent angiogenesis leading to mCP vascularization.

## Developmental relationships between meningeal FGPs and mCP vasculature

A previous study indicates that zebrafish FGPs regulate angiogenesis in the brain meningeal compartment (*Bower et al., 2017*). Since the DLV and PCeV develop in close proximity to the dorsal surfaces of the brain meninges, we examined developmental relationships between FGPs and mCP vasculature (*Figure 3—figure supplement 1*). To assess a role for FGPs in mCP vascular development, we analyzed *vegfc*[-/-];*vegfd*[-/-] larvae at 120 hpf since FGPs require Vegfc/Vegfd signaling to

migrate to the optic tectum by this stage (*Bower et al., 2017*; *van Lessen et al., 2017*; *Venero Galanternik et al., 2017*). To assess a role for mCP vasculature in FGP development, we analyzed 120 hpf *vegfab*$^{-/-}$ larvae which lacked the DLV. We found that the *vegfab*$^{-/-}$ fish lacking the DLV all formed FGPs similarly to their WT sibling controls, while *vegfc*$^{-/-}$;*vegfd*$^{-/-}$ larvae, which lacked FGPs, formed the DLV and PCeV (*Figure 3—figure supplement 1*). Consistent with these *vegfc*$^{-/-}$; *vegfd*$^{-/-}$ results, the *ccbe1* RNPs-injected larvae completely lacked FGPs, but formed the DLV and PCeV, at 120 hpf (*Figure 4C,D*). These observations demonstrate that the formation of FGPs and mCP vasculature is an independent developmental process. The recently discovered intracranial lymphatic vessels in zebrafish begin to form at much later stages of development (9–10 dpf) (*Castranova et al., 2021*), thus mCP vascularization processes should not be affected by these intracranial lymphatics.

## Vegfab/Vegfc- and Vegfab/Vegfd-dependent angiogenesis in the formation of other vascular beds

To address the specificity and requirements for Vegfab/Vegfc- and Vegfab/Vegfd-dependent angiogenic signaling in the formation of other vascular beds, we examined intersegmental vessel (ISV) formation in the trunk of these double mutants at 96 hpf (*Figure 2H'–M'*). We chose 96 hpf for this analysis because ISV formation is complete before 72 hpf (*Isogai et al., 2001*), and this was the developmental stage at which we noted severe defects in DLV and PCeV formation in these mutants (*Figure 2F,G*). We observed no significant differences in the number of ISVs between WT and any of the single or double mutant groups (*Figure 2N*). These data provide further evidence that Vegfab/Vegfc- and Vegfab/Vegfd-dependent angiogenic signaling is specifically required to drive mCP vascularization.

## Analysis of vEC molecular markers for the fenestrated and BBB states

We next asked how genetic elimination of these Vegf ligands affects the fenestrated state of the mCP vasculature. To this end, we analyzed *Tg(plvap:*EGFP) and *Tg(glut1b:*mCherry) reporter expression in *vegfab*, *vegfc*, and *vegfd* single mutants (*Figure 5*). At 10 dpf, WT fish exhibited strong *Tg(plvap:*EGFP) and faint *Tg(glut1b:*mCherry) reporter expression in the mCP vasculature, including in the DLV and PCeV (*Figure 5A–A"*). We observed that none of these single mutants displayed an obvious difference in the expression patterns of *Tg(plvap:EGFP)* and *Tg(glut1b:mCherry)* reporters in these vessels (*Figure 5B–D"*) as compared to WT fish, suggesting that the single mutants that formed the DLV and PCeV retain the fenestrated state of mCP vasculature.

Next, we examined Claudin-5 BBB marker expression in *vegfab*, *vegfc*, and *vegfd* single mutants and in combined mutants. We expected that if a loss of either of these Vegf ligands leads to a conversion of the mCP vasculature from the fenestrated to the BBB state, increased Claudin-5 expression should be observed in these vessels. In 10 dpf WT fish, Claudin-5 immunoreactivity is prominent in mCP epithelial cells and weakly present in the tissues located ventrally to the DLV and PCeV but is barely detectable in the DLV and PCeV (*Figure 5—figure supplement 1A–A"*). However, this low level of Claudin-5 expression in these vessels was not changed in the 10 dpf *vegfab*$^{-/-}$, *vegfab*$^{-/-}$; *vegfc*$^{-/-}$, *vegfab*$^{-/-}$;*vegfd*$^{-/-}$ and *vegfc*$^{-/-}$;*vegfd*$^{-/-}$ larvae that formed these vessels (*Figure 5—figure supplement 1B–D"*; data not shown). These results imply that the fenestrated endothelial state of the mCP vasculature is not altered in the absence of these Vegf ligands.

## Expression patterns of *vegfab*, *vegfc*, and *vegfd* during mCP vascularization

To determine cellular mechanisms governing mCP vascularization, we examined the expression patterns of *vegfab*, *vegfc* and *vegfd*. To achieve this aim at the single-cell resolution, we generated new BAC transgenic reporter lines, each of which drives the expression of Gal4FF under the control of a *vegfab*, *vegfc*, or *vegfd* promoter in a BAC. We crossed each of these lines with *Tg(UAS:EGFP)* fish, which carry an EGFP gene downstream of the upstream activating sequence (UAS). We validated these BAC transgenic lines by observing their EGFP expression domains, which closely matched respective mRNA expression patterns (*Figure 6—figure supplement 1A–F'*). Thus, these BAC transgenic tools reliably reflect the endogenous expression patterns of *vegfab*, *vegfc*, and *vegfd*.

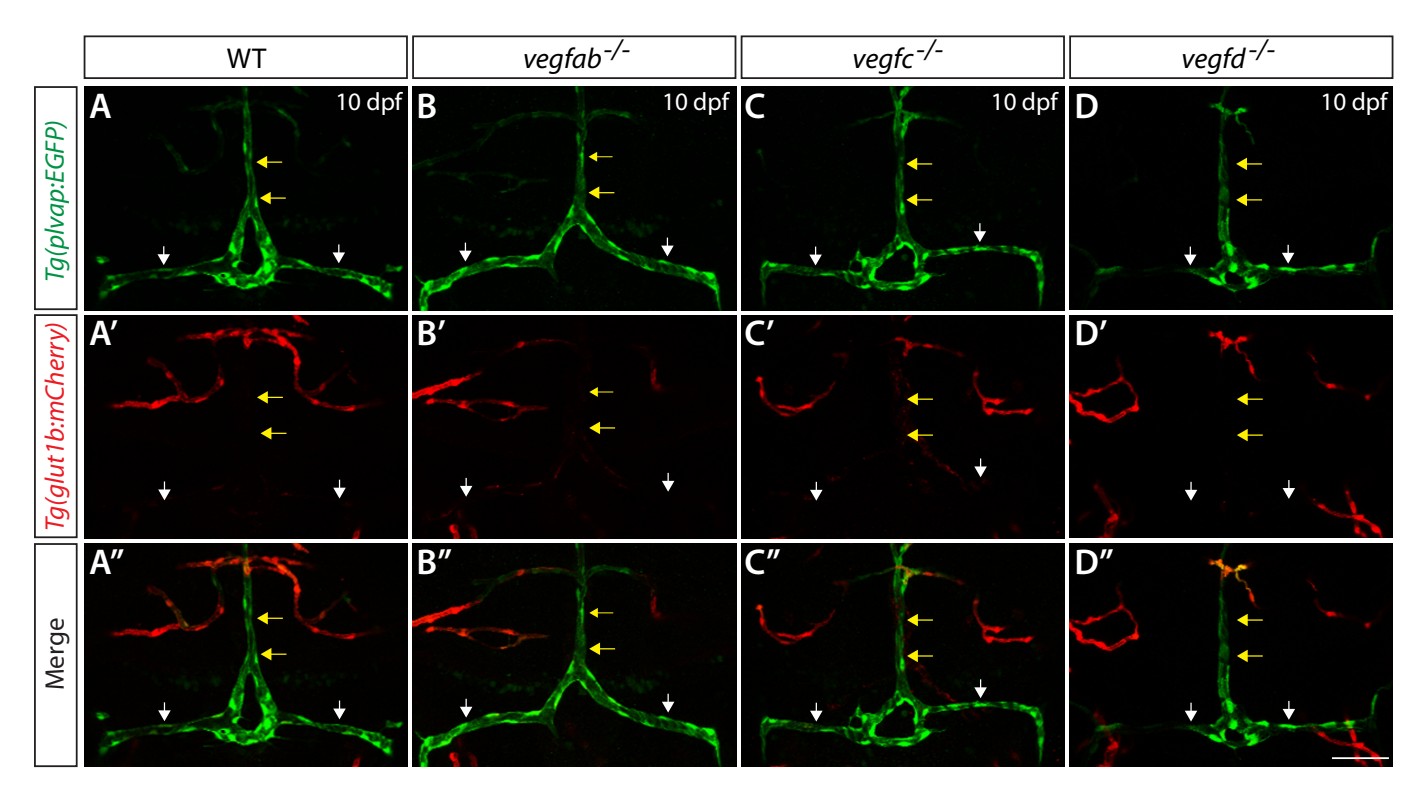

**Figure 5.** Expression analysis of endothelial markers for the fenestrated and BBB states in WT, *vegfab*, *vegfc*, and *vegfd* mutants. Dorsal views of 10 dpf WT (A–A"), *vegfab*^-/- (B–B"), *vegfc*^-/- (C–C"), and *vegfd*^-/- (D–D") heads of the larvae that carried the *Tg(plvap:EGFP)* (A–D) and *Tg(glut1b:mCherry)* (A'–D') transgenes (merged images shown in A"–D"). Only the *vegfab*^-/-, *vegfc*^-/-, and *vegfd*^-/- larvae that formed the DLV and PCeV were subjected to this analysis. High levels of *Tg(plvap*:EGFP), but low levels of *Tg(glut1b*:mCherry), expression were observed in the DLV and PCeV of WT fish (A–A', n = 24). The expression patterns of the *Tg(plvap:EGFP)* and *Tg(glut1b:mCherry)* transgenes were not altered in any of the single mutants (n = 9 for *vegfab*^-/-, n = 4 for *vegfc*^-/-, and n = 4 for *vegfd*^-/- fish). Yellow arrows point to the DLV and white arrows to the PCeV. Scale bar: 50 μm.

The online version of this article includes the following figure supplement(s) for figure 5:

**Figure supplement 1.** Analysis of Claudin-5 protein expression in the mCP vasculature of *vegf* single and double mutants.

To analyze the morphology of individual cells labeled by these BAC lines, we crossed each of them with *Tg(UAS:EGFP-CAAX)* fish, which carry a membrane-bound EGFP gene downstream of UAS. These generated double Tg fish are abbreviated *TgBAC(vegfab:EGFP)*, *TgBAC(vegfc:EGFP)*, and *TgBAC(vegfd:EGFP)* and were crossed with the endothelial cell-specific *Tg(kdrl:ras-mCherry)* line for expression analyses.

During the developmental stages at which the DLV extends toward the mCP and branches to join the PCeV (48–72 hpf), we observed that *vegfab* and *vegfc* are expressed in multiple cell types along the DLV migratory route (*Figure 6A–D"* and *Figure 6—figure supplement 1G–I, K, L*). We identified two prominent regions of *vegfab* expression: (1) meningeal fibroblast-like cells near the dorsal midline junction (DMJ) where DLV sprouting initiates and (2) mCP cells where the DLV and PCeV join to form the mCP vasculature (*Figure 6A–B"* and *Figure 6—figure supplement 1H,K,L*). The latter's reporter expression marked Claudin-5^+ mCP epithelial cells at 72 hpf (*Figure 6G–G"*). We observed *vegfc* expression in (1) meningeal fibroblast-like cells near the DMJ and (2) in the migrating vECs, including the growing tip and trailing cells, which comprise the DLV (*Figure 6C–D"* and *Figure 6—figure supplement 1I*). We also observed strong *vegfd* expression restricted to meningeal fibroblast-like cells at the DMJ and other regions, most of which are located anterior to growing DLV tip cells (*Figure 6E–F"* and *Figure 6—figure supplement 1J*). These observations suggest that Vegfab and Vegfd control mCP vascular formation via paracrine mechanisms, while Vegfc acts as a paracrine and/or autocrine factor to regulate this process.

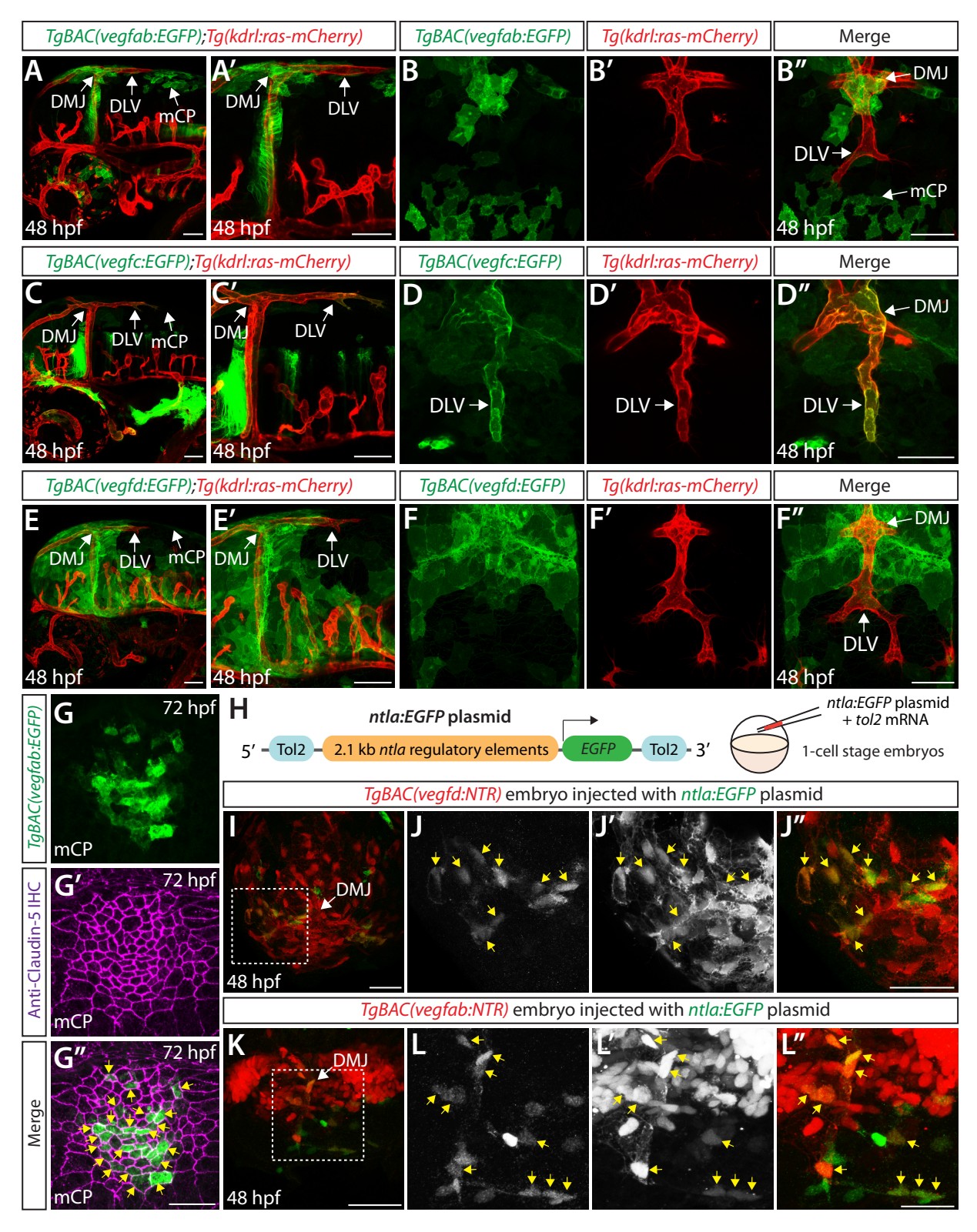

**Figure 6.** Expression analyses of *vegfab*, *vegfc*, and *vegfd* during mCP vascularization. (**A, A'** and **B–B"**) Lateral (**A and A'**) and dorsal (**B–B"**) views of 48 hpf *TgBAC(vegfab:EGFP);Tg(kdrl:ras-mCherry)* embryos, showing *TgBAC(vegfab:*EGFP) expression in the dorsal midline junction (DMJ) and mCP regions. *TgBAC(vegfab:*EGFP) expression was not observed in the extending DLV (**B–B"**). (**C, C'** and **D–D"**) Lateral (**C and C'**) and dorsal (**D–D"**) views of 48 hpf *TgBAC(vegfc:EGFP);Tg(kdrl:ras-mCherry)* embryos, showing *TgBAC(vegfc:*EGFP) expression in the endothelial cells that comprise the DLV (**D–**

*Figure 6 continued on next page*

*Figure 6 continued*

D"). (**E, E'** and **F–F"**) Lateral (**E** and **E'**) and dorsal (**F–F"**) views of 48 hpf *TgBAC(vegfd:EGFP);Tg(kdrl:ras-mCherry)* embryos, showing *TgBAC(vegfd:* EGFP) expression in the meninges and DMJ. *TgBAC(vegfd:*EGFP) expression was not observed in the extending DLV (**F–F"**). (**G–G"**) Dorsal views of a 72 hpf *TgBAC(vegfab:EGFP)* larva immunostained for Claudin-5 (magenta). Magnified images of the mCP region are shown. Yellow arrows indicate EGFP$^+$ and Claudin-5$^+$ mCP epithelial cells. (**H**) Schematic of the *ntla:EGFP* construct used for injection experiments (**I–L"**). (**I–L"**) Dorsal views of 48 hpf *TgBAC(vegfd:NTR)* (**I–J"**) and *TgBAC(vegfab:NTR)* (**K–L"**) head of embryos that were injected with the *ntla:EGFP* construct at the one-cell stage. Magnified images of the boxed area in (**I**) and (**K**) are shown in (**J–J"**) and (**L–L"**), respectively. Most of the EGFP$^+$ cells (**J and L**) were co-localized with NTR-mCherry$^+$ meningeal cells (**J' and L'**) in *TgBAC(vegfd:NTR)* (**J–J"**) and *TgBAC(vegfab:NTR)* (**L–L"**) embryos. Yellow arrows point to the co-localized cells (n = 3 embryos, 18 co-localized cells observed for *TgBAC(vegfd:NTR)*; n = 3 embryos, 19 co-localized cells observed for *TgBAC(vegfab:NTR)*). Scale bars: 50 µm in (**A–F"**), (**I**), (**J"**), (**K**); 25 µm in (**G", L"**).

The online version of this article includes the following figure supplement(s) for figure 6:

**Figure supplement 1.** Validation and expression analyses of distinct *vegf* BAC transgenic lines.

**Figure supplement 2.** *TgBAC(vegfd:*EGFP) and *TgBAC(vegfc:*EGFP) expression patterns in WT and *vegfab* mutants which displayed the presence or absence of the DLV.

**Figure supplement 3.** Co-localization analyses in *TgBAC(vegfab:NTR)*, *TgBAC(vegfc:NTR)*, and *TgBAC(vegfd:NTR)* embryos that carried the *Tg(sox10: EGFP)* transgene.

**Figure supplement 4.** Co-localization analyses in *TgBAC(vegfab:NTR)*, *TgBAC(vegfc:NTR)*, and *TgBAC(vegfd:NTR)* embryos carrying the *Tg(mpeg1.1: Dendra2)* transgene.

**Figure supplement 5.** Co-localization of the mesodermal cell marker and *TgBAC(vegfc:NTR)*$^+$ meningeal cells, and close similarity in cell morphology between mesoderm-derived meningeal fibroblasts and *vegf*-expressing meningeal cells.

Since our *vegfab*, *vegfc*, and *vegfd* expression analyses revealed their distinct and partially overlapping expression patterns, we investigated whether the deleterious mutations leading to a loss of Vegfab, Vegfc, or Vegfd functional proteins trigger transcriptional adaptation responses, a mechanism of genetic compensation that leads to the transcriptional modulation of related genes (*El-Brolosy et al., 2019*; *Sztal and Stainier, 2020*). For this purpose, we utilized the *TgBAC(vegfc: EGFP)* and *TgBAC(vegfd:EGFP)* lines to compare their expression patterns between *vegfab$^{+/+}$* and *vegfab$^{-/-}$* embryos generated from *vegfab$^{+/-}$* incrosses (*Figure 6—figure supplement 2*). We analyzed these embryos at 48 or 54 hpf – the developmental stages at which the DLV and PCeV are forming. We also compared the expression patterns of *vegfab$^{-/-}$* embryos that exhibited the presence, or absence, of the DLV. However, we did not find any obvious difference in *TgBAC(vegfc:* EGFP) and *TgBAC(vegfd:*EGFP) expression patterns between any of the *vegfab* genotypes and also between the *vegfab$^{-/-}$* embryos that displayed the presence, or absence, of the DLV (*Figure 6—figure supplement 2*). These observations indicate that although Vegfab, Vegfc, and Vegfd are functionally redundant in driving mCP vascularization, transcriptional adaptation may not be involved in this redundancy or mutations in only one of these *vegf* genes may not be sufficient to trigger transcriptional adaptation responses at the stages of mCP vessel development.

## DLV formation requires vEC-autonomous Vegfc signaling and paracrine Vegfab/Vegfd signaling

To determine the role of Vegfc as a paracrine and/or autocrine factor, we analyzed the separate *vegfc* mutation, *vegfc$^{um18}$*, which was previously isolated from a forward genetic screen (*Villefranc et al., 2013*). This particular mutation was shown to generate a prematurely truncated Vegfc protein that lacks efficient secretory and paracrine activity but retains the ability to activate its Flt4 receptor (in another words, cell-autonomous activity of Vegfc is retained) (*Villefranc et al., 2013*). Interestingly, we found that the homozygous embryos harboring this particular mutation did not exhibit a defect in DLV formation at 54 hpf (*Figure 7A–D*). This result is in contrast with that observed in the homozygous embryos of the same age that carry the *vegfc$^{hu6410}$* mutation which causes the early truncation of the Vegfc protein and leads to a deficiency in both paracrine and autocrine activities, resulting in the 'No DLV' phenotype (*Figure 1N,P*). These findings suggest that vEC-autonomous Vegfc function is important for DLV formation.

Since we found that *vegfab*, *vegfc*, and *vegfd* are all expressed in meningeal fibroblast-like cells, we next sought to address the contribution of this cell type to mCP vascularization. Meningeal fibroblasts were previously shown to derive from both neural crest cells and the mesoderm (*Siegenthaler and Pleasure, 2011*). To determine the identity of the meningeal fibroblast-like cells,

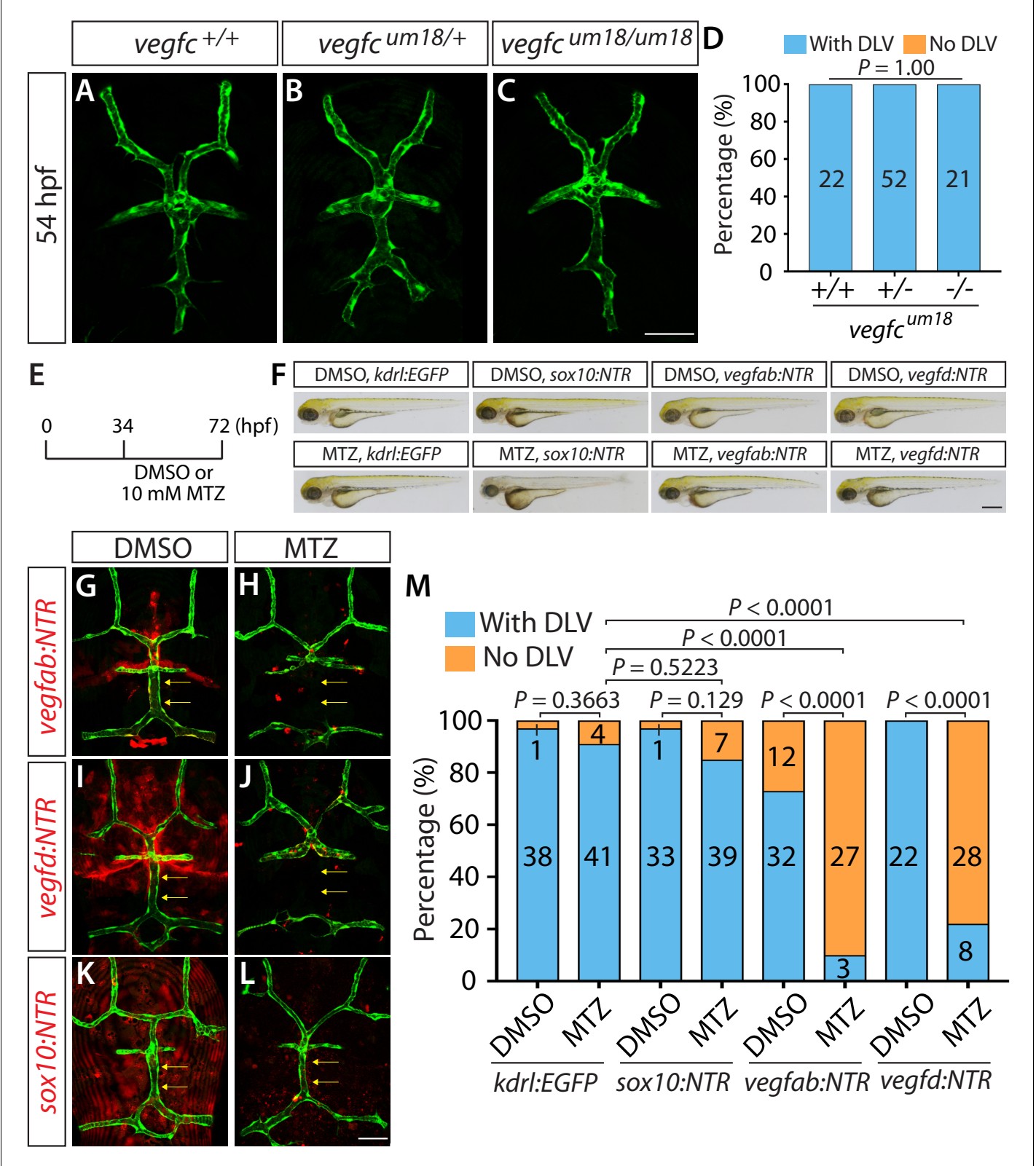

**Figure 7.** Endothelial-autonomous Vegfc and meningeal fibroblast-derived Vegfab and Vegfd are important for DLV formation. (A–C) Dorsal views of 54 hpf *vegfc⁺/⁺* (A), *vegfcᵘᵐ¹⁸/⁺* (B), and *vegfcᵘᵐ¹⁸/ᵘᵐ¹⁸* (C) cranial vasculature visualized by *Tg(kdrl:EGFP)* expression. *vegfcᵘᵐ¹⁸/ᵘᵐ¹⁸* fish did not exhibit a defect in DLV formation (C) in contrast to the 'No DLV' phenotype found in *vegfcʰᵘ⁶⁴¹⁰/ʰᵘ⁶⁴¹⁰* fish (**Figure 1N**). (D) Percentage of 54 hpf fish of indicated genotype with and without the DLV (n = 22 for *vegfc⁺/⁺*, n = 52 for *vegfcᵘᵐ¹⁸/⁺*, and n = 21 for *vegfcᵘᵐ¹⁸/ᵘᵐ¹⁸* fish). (E) Time course of the cell

*Figure 7 continued on next page*

**Figure 7 continued**

ablation experiments for panels (**F**–**M**). (**F**) Brightfield images of 72 hpf *Tg(kdrl:EGFP)*, *Tg(sox10:NTR)*, *TgBAC(vegfab:NTR)*, and *TgBAC(vegfd:NTR)* larvae after treatment with DMSO or 10 mM Metronidazole (MTZ) from 34 to 72 hpf. (**G**–**L**) Dorsal views of 72 hpf *Tg(kdrl:EGFP);TgBAC(vegfab:NTR)* (**G** and **H**), *Tg(kdrl:EGFP);TgBAC(vegfd:NTR)* (**I** and **J**), and *Tg(kdrl:EGFP);Tg(sox10:NTR)* (**K** and **L**) cranial vasculature after treatment with DMSO (**G**, **I**, **K**) or MTZ (**H**, **J**, **L**) from 34 to 72 hpf. Efficient ablation of NTR-mCherry$^+$ cells was observed after treatment with MTZ (**H**, **J**, **L**) compared to their respective DMSO-treated controls (**G**, **I**, **K**). MTZ-treated *TgBAC(vegfab:NTR)* and *TgBAC(vegfd:NTR)*, but not *Tg(sox10:NTR)*, larvae lacked the DLV (arrows). (**M**) Percentage of the indicated 72 hpf transgenic fish with and without the DLV after treatment with DMSO or MTZ (the number of the animals examined per each treatment group is listed in the graph). MTZ-treated *TgBAC(vegfab:NTR)* and *TgBAC(vegfd:NTR)* larvae exhibited a drastic increase in the 'No DLV' phenotype when compared to their corresponding DMSO treatment groups and the MTZ-treated *Tg(kdrl:EGFP)* larval group. In contrast, MTZ-treated *Tg(kdrl:EGFP)* and *Tg(sox10:NTR)* animals showed no significant differences compared to their corresponding DMSO treatment groups. Scale bars: 1 mm in (**F**); 50 μm in (**C** and **L**).

we used the *Tg(sox10:EGFP)* line, which labels the cells derived from the neural crest. For this purpose, we crossed each of the BAC Tg lines with *Tg(UAS:NTR-mCherry)* fish, which carry a *Nitroreductase-mCherry* (*NTR-mCherry*) fusion gene downstream of UAS. These resulting double Tg fish are abbreviated *TgBAC(vegfab:NTR)*, *TgBAC(vegfc:NTR)*, and *TgBAC(vegfd:NTR)*. We found that a vast majority (over 96%) of the NTR-mCherry expression driven by each of the BAC lines did not co-localize with the *Tg(sox10:*EGFP) expression in the dorsal surfaces of the meninges, including at the DMJ at 48 hpf (*Figure 6—figure supplement 3*). However, we noted that the *Tg(sox10:*EGFP)$^+$ cells and NTR-mCherry$^+$ fibroblast-like cells marked by each of these BAC lines were present in the meninges in an intermingled manner as observed in mesoderm-specific *Tg(ntla:Gal4);Tg(Kaede)* embryos crossed with the *Tg(sox10:mRFP)* neural crest line (*Figure 6—figure supplement 3A–A''*), suggesting that the meningeal fibroblast-like cells are likely meningeal fibroblasts of mesodermal origin.

Immune cells, including macrophages and neutrophils, are also known to reside in brain meningeal compartments; however, we observed that meningeal *Tg(mpeg1.1:*Dendra2)$^+$ macrophages were not co-localized with the NTR-mCherry$^+$ cells marked by each of the BAC lines (*Figure 6—figure supplement 4*) and that *Tg(lyz:*EGFP)$^+$ neutrophils do not reside in the dorsal surfaces of the brain meninges at 48 hpf, including in the regions near the DMJ and mCP (data not shown).

To investigate whether mesoderm-derived meningeal fibroblasts are a Vegfs-expressing cell type, we engineered a plasmid in which EGFP expression is driven under the control of the putative zebrafish *ntla* promoter, the 2.1 kb regulatory elements used previously to generate the *Tg(ntla:Gal4)* line (*Figure 6H*; *Harvey et al., 2010*; *Lee et al., 2013*). When we injected this plasmid at the one-cell stage, we observed sparsely labeled EGFP$^+$ cells in the dorsal brain meninges that were mostly co-localized (over 82%) with the NTR-mCherry$^+$ cells in *Tg(ntla:Gal4);Tg(UAS:NTR-mCherry)* embryos (*Figure 6—figure supplement 5B–C''*). Thus, this plasmid injection can reliably label mesoderm-derived cell lineages, including meningeal fibroblasts and vECs, in a mosaic fashion. Next, we injected the same plasmid into *TgBAC(vegfab:NTR)*, *TgBAC(vegfc:NTR)*, or *TgBAC(vegfd:NTR)* one-cell stage embryos. We observed that the NTR-mCherry$^+$ cells marked by each of the BAC lines were also co-localized with the EGFP$^+$ cells in the meninges, including in the region near the DMJ (*Figure 6I–L"* and *Figure 6—figure supplement 5D–E''*). Moreover, we observed close similarities in cell morphology between the meningeal cells marked by *TgBAC(vegfd:EGFP)*, *TgBAC(vegfab:EGFP)*, and the *Tg(ntla:Gal4)* embryos carrying the *Tg(UAS:EGFP-CAAX)* transgene (*Figure 6—figure supplement 5F–H*). These results provide evidence that mesoderm-derived meningeal fibroblasts are a source of these Vegf ligands in the dorsal meningeal compartments of the brain.

To determine the function of the meningeal fibroblasts labeled by *vegfab* and *vegfd* BAC reporter expression, we eliminated these cell types using an established chemogenetic ablation approach in zebrafish that utilizes the bacterial enzyme Nitroreductase (NTR) to induce apoptotic cell death upon administration of its substrate prodrug metronidazole (MTZ) (*Curado et al., 2007*). This approach allows target cell ablation in a spatiotemporally controlled manner. Since a vast majority of the meningeal fibroblast-like cells labeled by these BAC Tg lines were negative for *Tg(sox10:*EGFP) expression, we generated *Tg(sox10:gal4VP-16);Tg(UAS:NTR-mCherry)*, abbreviated *Tg(sox10:NTR)*, animals to ablate neural crest-derived meningeal fibroblasts for comparison. To minimize toxic effects of cell ablation during early embryogenesis, we began treating *TgBAC(vegfab:NTR)*, *TgBAC(vegfd:NTR)*, and *Tg(sox10:NTR)* embryos with 10 mM MTZ starting at 34 hpf and analyzed them at

72 hpf (*Figure 7E, F*). We also treated *Tg(kdrl:EGFP)* embryos with 10 mM MTZ or DMSO as controls to assess the effect of MTZ treatment itself on DLV formation. We validated that these NTR lines and the time course we employed induced the efficient ablation of target cells upon MTZ administration (*Figure 7G–L*). We observed that the MTZ-treated *Tg(kdrl:EGFP)* and *Tg(sox10:NTR)* larvae did not show any significant difference in the percentage of the 'No DLV' phenotype (9% and 15%, respectively) when compared to their respective DMSO-treated groups (3% and 3%, respectively) (*Figure 7M*). However, we found that the MTZ-treated *TgBAC(vegfab:NTR)* and *TgBAC (vegfd:NTR)* larvae exhibited a pronounced increase in this phenotype (90% and 78%, respectively) when compared to their corresponding DMSO-treated groups (27% and 0%, respectively) as well as the MTZ-treated *Tg(kdrl:EGFP)* group (9%) (*Figure 7M*). Given our observations that *vegfab* BAC reporter expression marked mesoderm-derived meningeal fibroblasts and mCP epithelial cells while *vegfd* BAC reporter expression marked only mesoderm-derived meningeal fibroblasts around the DMJ and mCP regions (*Figure 6A, B", E and F"*), these findings suggest that mesoderm-derived meningeal fibroblasts are an important source of these Vegf ligands necessary for DLV formation. Notably, these results also suggest that neural crest-derived meningeal fibroblasts are not crucial for this process, indicating that meningeal fibroblasts of distinct origins differentially contribute to mCP vascularization.

## Expression patterns of *vegfaa* and its requirement for mCP vascular formation

Since Vegfaa plays a key role in angiogenesis during early embryogenesis in zebrafish (*Rossi et al., 2016*), we next asked what role Vegfaa plays in mCP vascular development. *vegfaa* mutants are known to exhibit severe early vascular defects (*Rossi et al., 2016*), which makes it difficult to precisely assess a role for Vegfaa in angiogenesis at later developmental stages. To overcome this technical limitation, we generated a new BAC transgenic reporter line, *TgBAC(vegfaa:gal4ff)*, which drives the expression of Gal4FF under the control of a *vegfaa* promoter in a BAC. By crossing this line with *Tg(UAS:EGFP)* fish, we were able to observe that the EGFP expression domains driven by this BAC transgenic line closely matched the endogenous mRNA expression patterns of *vegfaa* (*Figure 8A–B'*). Thus, this BAC transgenic tool reliably reflects the endogenous expression patterns of *vegfaa*.

We next generated *TgBAC(vegfaa:gal4ff);Tg(UAS:EGFP-CAAX)* fish, abbreviated *TgBAC(vegfaa: EGFP)*, to examine *vegfaa* expression patterns at the single-cell resolution as we identified the cell types that express *vegfab*, *vegfc*, and *vegfd* using their BAC lines. Intriguingly, in contrast to the *vegfab*, *vegfc*, and *vegfd* BAC reporter expression, we did not observe EGFP$^+$ meningeal fibroblast-like cells near the DMJ at 48 hpf in *TgBAC(vegfaa:EGFP)* embryos (*Figure 8C–D"*). *TgBAC(vegfaa: EGFP)$^+$* cells were present in the brain parenchyma ventral to the DMJ, but not in the dorsal meningeal compartments of the brain (*Figure 8C–D"*).

Next, we generated *TgBAC(vegfaa:gal4ff);Tg(UAS:NTR-mCherry)* fish, abbreviated *TgBAC(vegfaa:NTR)*, to ablate Vegfaa-expressing cells in a temporally controlled manner, which allowed us to analyze potential Vegfaa function by minimizing early vascular defects caused by lack of Vegfaa function. To this end, we treated *TgBAC(vegfaa:NTR)* with 10 mM MTZ or DMSO from 34 to 72 hpf (*Figure 8E*) – exactly the same experimental time course we used to conduct the cell ablation shown in *Figure 7E–M*. Under this cell ablation condition, MTZ-treated *TgBAC(vegfaa:NTR)* fish developed a slight edema (*Figure 8F,G*); however, a vast majority of them (95%) formed the DLV and PCeV (*Figure 8I,J*) similarly to the DMSO-treated control siblings (93%) (*Figure 8H,J*). These expression and cell ablation results suggest that Vegfaa and Vegfaa-expressing cells are not involved in DLV and PCeV formation, supporting a model in which Vegfab/Vegfc/Vegfd redundant actions are central to this process.

## Discussion

Here, we provide the first evidence for molecular cues and cell types responsible for driving the development of fenestrated CP vasculature in vertebrates. The present study also provides the first ultrastructural information about a zebrafish larval CP and its epithelial cells, which illustrates the conservation of the CPs across vertebrate species at the ultrastructural level. By conducting thorough phenotypic characterization of single, double, and triple loss-of-function zebrafish *vegf*

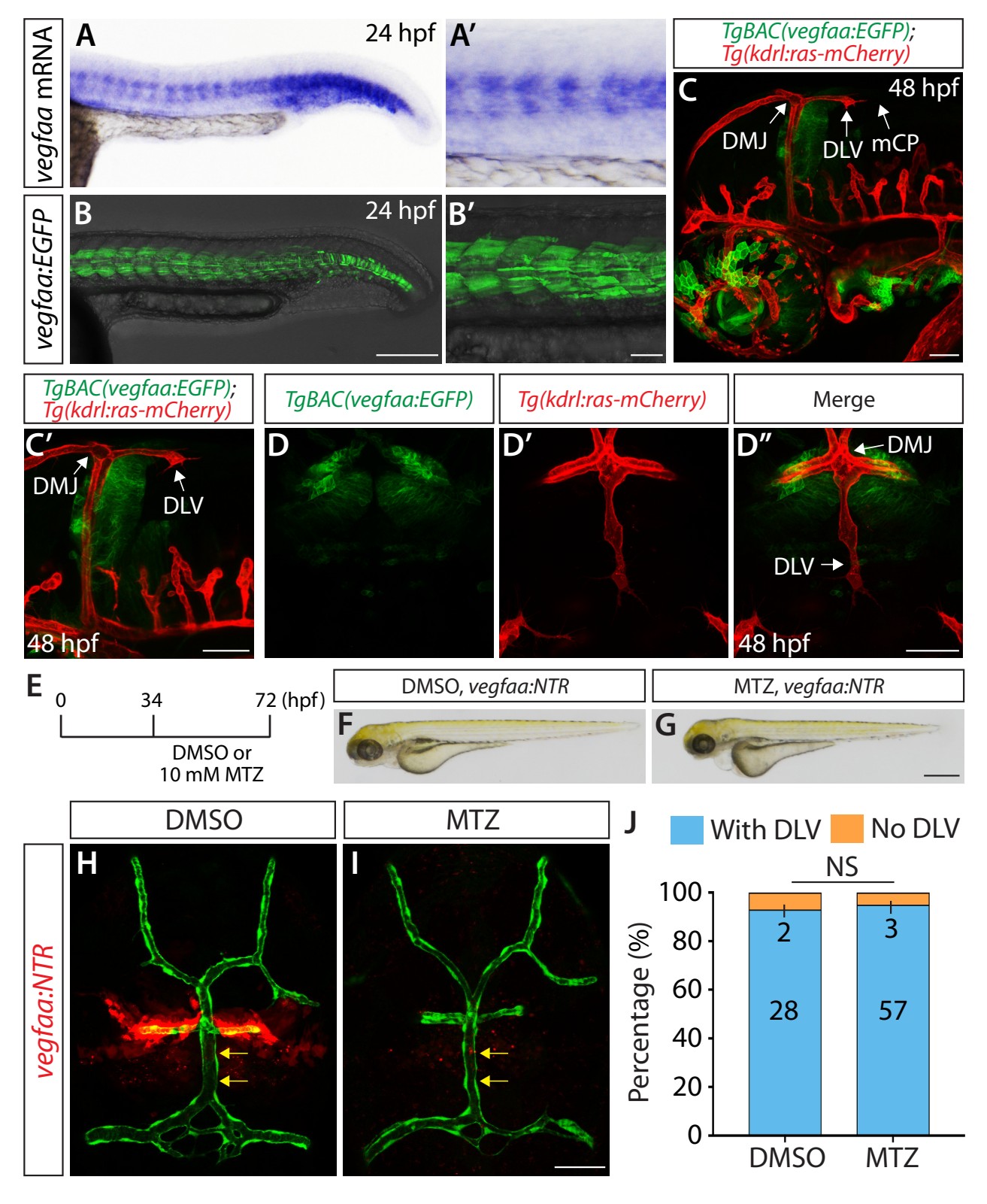

**Figure 8.** Expression patterns of *vegfaa* and its requirement for mCP vascular formation. (A–B') Lateral views of endogenous *vegfaa* mRNA expression patterns detected by in situ hybridization at 24 hpf (**A** and **A'**). Lateral views of the EGFP expression patterns driven by the *Tg(UAS:EGFP)* embryo carrying the *TgBAC(vegfaa:gal4ff)* transgene (**B** and **B'**). The EGFP expression domains closely matched the endogenous *vegfaa* mRNA expression patterns. (**C, C'** and **D–D"**) Lateral (**C** and **C'**) and dorsal (**D–D"**) views of 48 hpf *TgBAC(vegfaa:EGFP);Tg(kdrl:ras-mCherry)* embryos, showing the
*Figure 8 continued on next page*

*Figure 8 continued*

*TgBAC(vegfaa*:EGFP) expression in the cells located ventrally to the DLV. *TgBAC(vegfaa*:EGFP) expression was not observed in the extending DLV or in the dorsal surfaces of the brain meninges. (E) Time course of the cell ablation experiments for panels (F–J). (F and G) Brightfield images of 72 hpf *TgBAC(vegfaa*:NTR) larvae after treatment with DMSO (F) or 10 mM MTZ (G) from 34 to 72 hpf. (H and I) Dorsal views of 72 hpf *TgBAC(vegfaa*:NTR) cranial vasculature visualized by *Tg(kdrl*:EGFP) expression after treatment with DMSO (H) or MTZ (I) from 34 to 72 hpf. Efficient ablation of NTR-mCherry$^+$ cells was observed after treatment with MTZ (I) compared to the DMSO-treated control (H). MTZ-treated *TgBAC(vegfaa*:NTR) larvae formed the DLV (yellow arrows) and PCeV. (J) Percentage of the 72 hpf transgenic fish with and without the DLV after treatment with DMSO or MTZ. No significant difference was observed between MTZ-treated (n = 60) and DMSO-treated (n = 30) groups. Scale bars: 1 mm in (G); 200 μm in (B); 100 μm in (B'); 50 μm in (C), (C'), (D' and I).

mutants, we demonstrate that the functional redundancy among Vegfab, Vegfc, and Vegfd is key to driving fenestrated mCP vessel development since the individual mutants exhibited only mild or no defects in mCP vascular development. Additionally, we provide evidence that *ccbe1* and *vegfab* genetically interacts in mCP vascularization, identifying a novel role for Ccbe1 in cerebrovascular development, specifically in fenestrated CP vessel formation. Moreover, we identify cell types (vECs, mesoderm-derived meningeal fibroblasts, and mCP epithelial cells) that express *vegfab*, *vegfc*, and *vegfd* during mCP vascularization. These expression data combined with paracrine activity-deficient *vegfc*$^{um18}$ mutant analysis and cell-type-specific chemogenetic cell ablation support a model in which endothelial-autonomous Vegfc and paracrine actions of Vegfab/Vegfd from mesoderm-derived meningeal fibroblasts are critical for DLV formation. Altogether, these findings define a developmental mechanism critical for the generation of brain vascular heterogeneity, specifically fenestrated blood vessels.

## Combinations of angiogenic cues diversify region-specific, specialized vascular beds in the brain

Previous studies have shown that Wnt ligands (Wnt7a, Wnt7b, Norrin) and their receptors act in a partially redundant manner to direct the patterning and maintenance of BBB vascular beds in distinct regions of the CNS (*Cho et al., 2017*; *Wang et al., 2018*; *Zhou and Nathans, 2014*; *Zhou et al., 2014*). Combined losses of the Norrin and Wnt7a/Wnt7b signaling components were shown to result in far more severe defects in BBB-forming vessel patterning and maintenance in specific areas of the brain than the defects caused by the individual loss of these components (*Wang et al., 2018*; *Zhou and Nathans, 2014*). These results suggest a model in which combinations of canonical Wnt ligands and receptors direct BBB-forming vascular bed development in a CNS region-specific manner. This model is further supported by the recent reports showing that β-catenin activation downstream of the canonical Wnt signaling is maintained at strikingly lower levels in the fenestrated vascular beds of the CPs and circumventricular organs than in BBB-forming vascular beds (*Benz et al., 2019*; *Wang et al., 2019*). However, these prior studies have raised an important question as to what molecular cues direct the development of the fenestrated vascular beds in the CPs and circumventricular organs.

Our findings here identify a mechanism whereby redundant Vegf ligands selectively drive fenestrated CP vascular formation in the zebrafish brain, filling this gap in our scientific knowledge and raising new questions: (1) Do these Vegfs control vessel fenestration similarly to the canonical Wnt/β-catenin signaling that regulates both vessel formation and specializations in BBB-forming vascular beds?; (2) Does fenestrated vascular bed formation in the dCP or circumventricular organs require the same or different combinations of Vegf ligands compared to those critical for fenestrated mCP vessel development?; (3) How are these multiple Vegf ligand-induced signals transmitted by their receptors and downstream signaling pathways?; (4) How do these Vegfs achieve fenestrated mCP vessel-selective patterning with little effect on neighboring non-fenestrated vasculature?; and (5) What are upstream developmental events that determine the spatiotemporally restricted Vegf expression patterns? Addressing these questions will help to further understand similarities and differences in vEC's molecular requirements for angiogenesis leading to fenestrated and non-fenestrated brain vascular beds.

## Cellular mechanisms underlying mCP vessel-selective patterning by redundant Vegf ligands

How do multiple Vegfs functionally interact in directing mCP vascularization? Our expression analyses showed that mesoderm-derived meningeal fibroblasts are a common cell type that expresses *vegfab*, *vegfc*, and *vegfd* during mCP vascularization. In particular, *vegfd* expression is predominantly restricted to this cell type during this process, and ablation of Vegfd-expressing cells led to severely compromised DLV formation. Intriguingly, ablation of neural crest-derived meningeal fibroblasts did not result in impaired DLV formation. This result is consistent with our observations of these *vegfs'* expression restricted to mesoderm-derived, but not neural crest-derived, meningeal fibroblasts. We further found that *vegfaa* is not expressed in meningeal fibroblasts and that ablation of Vegfaa-expressing cells did not abrogate DLV formation. Collectively, these observations suggest that meningeal fibroblasts, specifically those derived from the mesoderm, are a key cell type which drives DLV formation and that the multiple Vegfs' functional redundancy likely occurs at the cell-type level.

Transcriptional adaptation is a recently described phenomenon involved in genetic compensation by which a mutation in one gene leads to the transcriptional modulation of related genes (*El-Brolosy et al., 2019*; *Sztal and Stainier, 2020*). To ask whether transcriptional adaptation explains the Vegfs' functional redundancy in this context, we examined *vegfc* and *vegfd* expression in *vegfab* mutants using their respective BAC transgenic lines. However, we did not find evidence of obvious changes in their expression patterns/levels in the dorsal surfaces of the brain meninges or in the regions near the DMJ and mCP, regardless of whether the *vegfab* mutants formed or failed to form the DLV. This result implies that functional redundancy among these Vegfs may not involve local transcriptional adaptation responses in these areas of the brain at the stage of DLV extension toward the mCP (48–54 hpf), although further investigations will be needed.

Ccbe1 is a highly conserved, secreted protein essential for proteolytic activation of Vegfc and lymphangiogenesis (*Bos et al., 2011*; *Hogan et al., 2009*; *Le Guen et al., 2014*). Intriguingly, combined genetic inactivation of *vegfab* and *vegfc* or *vegfab* and *ccbe1* displayed similarly increased penetrance and expressivity of DLV and PCeV defects, indicating that Ccbe1 is involved in mCP vascular development possibly through proteolytic activation of Vegfc. It will be important to further define Ccbe1-expressing cell type(s) and Ccbe1-mediated angiogenic regulation during mCP vascularization. Addressing these issues may explain how local, proteolytic activation of Vegf ligands mediated by molecules such as Ccbe1 contributes to the Vegfs-directed mCP vessel-selective patterning.

## Molecular mechanisms governing mCP vessel formation and fenestration

Do mCP vessel formation and fenestration rely on the same molecular cues? The mCP vasculature develops via the convergence of the DLV and PCeV. These converged vessels all exhibit the shared molecular signatures of the fenestrated state and become a functionally integrated vascular network. Almost all the *vegfab;vegfc;vegfd* triple mutants we examined lacked this entire network of the fenestrated vasculature. Although there is a possibility that the primary deficits in these mutants are the DLV/PCeV formation defects and that the lack of mCP vascularization is secondary to them, it is evident that mCP vessel formation requires the Vegfs-dependent angiogenesis.

In contrast, our expression analyses of several BBB and fenestration molecular markers in the *vegf* single or double mutants did not reveal obvious changes in Claudin-5 protein expression or *Tg*(*plvap*:EGFP);*Tg*(*glut1b*:mCherry) reporter expression in these vessels. Given the fact that these analyses were feasible only for the mutants that formed the DLV and PCeV, it is possible that compensatory mechanisms which allow the development of these vessels in the *vegf* mutants also restore fenestration marker expression in these vessels. Alternatively, these results may indicate that the formation and fenestration of the mCP vasculature are governed by separate mechanisms. In light of the recent reports in the field (*Anbalagan et al., 2018*; *Wang et al., 2019*), we speculate that vessel fenestration in the CP may require combined actions of (1) endogenous inhibitor(s) of canonical Wnt/β-catenin signaling; (2) vascular permeability/inflammatory protein(s); and/or (3) signaling molecules inhibiting endothelial tight junction formation. These combined actions may allow vECs in the CP to (1) maintain low levels of Wnt/β-catenin signaling, (2) exhibit increased vascular permeability, and (3) form loose endothelial connections. Further investigations into these or alternative possibilities will address how mCP vessel fenestration is developed and maintained at the molecular levels.

In summary, defining the unique sets of angiogenic factors and their receptors required for the development and maintenance of each vascular bed within the CNS is fundamental in order to understand the pathophysiology, malformations, and diseases of the CNS vasculature. Given the broad range of pathophysiological processes regulated by Vegf signaling, our findings may provide insights into the underlying mechanisms involved in pathophysiological angiogenesis, vEC heterogeneity in other organs and tumors, as well as the clinical applications of angiogenic and anti-angiogenic therapy. Therefore, our elucidation of cellular and molecular events critical for fenestrated CP vascular development has general implications for understanding the establishment of organ-specific vascular beds and endothelial heterogeneity, in particular how fenestrated endothelial fate and its vessel formation are directed during development.

# Materials and methods

## Key resources table

| Reagent type (species) or resource | Designation | Source or reference | Identifiers | Additional information |
|---|---|---|---|---|
| Genetic reagent (*Danio rerio*) | *Tg(kdrl: EGFP)*$^{s843}$ | *Jin et al., 2005* | ZFIN: s843 | |
| Genetic reagent (*Danio rerio*) | *Tg(kdrl:Has. HRAS-mcherry)*$^{s896}$ | *Chi et al., 2008* | ZFIN: s896 | |
| Genetic reagent (*Danio rerio*) | *Tg(UAS: EGFP)*$^{nkuasgfp1a}$ | *Asakawa et al., 2008* | ZFIN: nkuasgfp1a | |
| Genetic reagent (*Danio rerio*) | *Tg(UAS:EGFP-CAAX)*$^{m1230}$ | *Fernandes et al., 2012* | ZFIN: m1230 | |
| Genetic reagent (*Danio rerio*) | *Tg(UAS-E1b: NfsB-mCherry)*$^{c264}$ | *Davison et al., 2007* | ZFIN: c264 | |
| Genetic reagent (*Danio rerio*) | *Tg(UAS: Kaede)*$^{rk8}$ | *Hatta et al., 2006* | ZFIN: rk8 | |
| Genetic reagent (*Danio rerio*) | *Tg(hsp70l:sflt1, cryaa-cerulean)*$^{bns80}$ | *Matsuoka et al., 2016* | ZFIN: bns80 | |
| Genetic reagent (*Danio rerio*) | *Tg(hsp70l:sflt4, cryaa-cerulean)*$^{bns82}$ | *Matsuoka et al., 2016* | ZFIN: bns82 | |
| Genetic reagent (*Danio rerio*) | *Tg(−4.9sox10: EGFP)*$^{ba2}$ | *Carney et al., 2006* | ZFIN: ba2 | |
| Genetic reagent (*Danio rerio*) | *Tg(sox10: mRFP)*$^{vu234}$ | *Kucenas et al., 2008* | ZFIN: vu234 | |
| Genetic reagent (*Danio rerio*) | *Tg(sox10:Gal4-VP16)*$^{sq9}$ | *Lee et al., 2013* | ZFIN: sq9 | |
| Genetic reagent (*Danio rerio*) | *Tg(ntla:Gal4-VP16)*$^{sq12}$ | *Lee et al., 2013* | ZFIN: sq12 | |
| Genetic reagent (*Danio rerio*) | *TgBAC(vegfab: gal4ff)*$^{bns273}$ | *Mullapudi et al., 2019* | ZFIN: bns273 | |
| Genetic reagent (*Danio rerio*) | *Tg(glut1b: mCherry)*$^{sj1}$ | *Umans et al., 2017* | ZFIN: sj1 | |
| Genetic reagent (*Danio rerio*) | *Tg(plvapb: EGFP)*$^{sj3}$ | *Umans et al., 2017* | ZFIN: sj3 | |
| Genetic reagent (*Danio rerio*) | *Et(cp:EGFP)*$^{sj2}$ | *Henson et al., 2014* | ZFIN: sj2 | |
| Genetic reagent (*Danio rerio*) | *Tg(mpeg1.1: Dendra2)*$^{uwm12}$ | *Harvie et al., 2013* | ZFIN: uwm12 | |

*Continued on next page*

Continued

| Reagent type (species) or resource | Designation | Source or reference | Identifiers | Additional information |
|---|---|---|---|---|
| Genetic reagent (*Danio rerio*) | *Tg(lyz:EGFP)^nz117* | *Hall et al., 2007* | ZFIN: nz117 | |
| Genetic reagent (*Danio rerio*) | *Tg(lyve1: DsRed)^nz101* | *Okuda et al., 2012* | ZFIN: nz101 | |
| Genetic reagent (*Danio rerio*) | *vegfaa^bns1* | *Rossi et al., 2016* | ZFIN: bns1 | |
| Genetic reagent (*Danio rerio*) | *vegfab^bns92* | *Rossi et al., 2016* | ZFIN: bns92 | |
| Genetic reagent (*Danio rerio*) | *vegfc^hu6410* | *Helker et al., 2013* | ZFIN: hu6410 | |
| Genetic reagent (*Danio rerio*) | *vegfc^um18* | *Villefranc et al., 2013* | ZFIN: um18 | |
| Genetic reagent (*Danio rerio*) | *vegfd^bns257* | *Gancz et al., 2019* | ZFIN: bns257 | |
| Genetic reagent (*Danio rerio*) | *TgBAC(vegfc: gal4ff)^bns270* | This paper | ZFIN: bns270 | |
| Genetic reagent (*Danio rerio*) | *TgBAC(vegfd: gal4ff)^lri95* | This paper | ZFIN: lri95 | |
| Genetic reagent (*Danio rerio*) | *TgBAC(vegfaa: gal4ff)^lri96* | This paper | ZFIN: lri96 | |
| Antibody | anti-GFP (chicken polyclonal) | Aves Labs | Cat#: GFP-1010 | 1:1000 |
| Antibody | anti-DsRed (rabbit polyclonal) | Clontech Labs | Cat#: 632496 | 1:300 |
| Antibody | anti-Claudin 5 (mouse monoclonal) | Thermo Fisher Scientific | Cat#: 35–2500 | 1:500 |
| Sequence-based reagent | *ccbe1* crRNA1 | This paper | CRISPR RNA | TTCTCCTC TCGGAAAG TCCA |
| Sequence-based reagent | *ccbe1* crRNA2 | This paper | CRISPR RNA | TACCCGTGCG TAAAGTCCAC |
| Sequence-based reagent | *ccbe1* crRNA3 | This paper | CRISPR RNA | CTTCTGGAA TACAC TGACCC |
| Peptide, recombinant protein | Alt-R S.p. Cas9 Nuclease V3 | Integrated DNA Technologies | Cat#: 1081058 | |
| Commercial assay or kit | DIG RNA Labeling Kit (SP6/T7) | Millipore Sigma | Cat#: 11175025910 | |
| Commercial assay or kit | mMessage mMachine T3 Transcription Kit | Thermo Fisher Scientific | Cat#: AM1348 | |
| Commercial assay or kit | RNA Clean and Concentrator-5 Kit | Zymo Research | Cat#: R1013 | |
| Chemical compound, drug | Qdot 655 Streptavidin Conjugate | Thermo Fisher Scientific | Cat#: Q10123MP | |
| Chemical compound, drug | Metronidazole | Millipore Sigma | Cat#: M3761 | |

*Continued on next page*

*Continued*

| Reagent type (species) or resource | Designation | Source or reference | Identifiers | Additional information |
|---|---|---|---|---|
| Software, algorithm | LAS X Version 3.7.0.20979 | Leica | | |
| Software, algorithm | NIS-Elements BR Imaging Software Version 5.10.01 | Nikon | | |
| Software, algorithm | GraphPad Prism 8 | GraphPad Software | | |
| Software, algorithm | Adobe Photoshop CS6 | Adobe | | |
| Software, algorithm | Adobe Illustrator CS6 | Adobe | | |

## Zebrafish husbandry and strains

All zebrafish husbandry was performed under standard conditions in accordance with institutional and national ethical and animal welfare guidelines. All zebrafish work was approved by the Cleveland Clinic's Institutional Animal Care and Use Committee under the protocol number 2018–1970. The following lines were used in this study: *Tg(kdrl:EGFP)s843* (*Jin et al., 2005*); *Tg(kdrl:Has.HRAS-mcherry)s896* (*Chi et al., 2008*), abbreviated *Tg(kdrl:ras-mCherry)*; *Tg(UAS:EGFP)nkuasgfp1a* (*Asakawa et al., 2008*); *Tg(UAS:EGFP-CAAX)m1230* (*Fernandes et al., 2012*); *Tg(UAS-E1b:NfsB-mCherry)c264* (*Davison et al., 2007*), abbreviated *Tg(UAS:NTR-mCherry)*; *Tg(UAS:Kaede)rk8* (*Hatta et al., 2006*); *Tg(hsp70l:sflt1, cryaa-cerulean)bns80* (*Matsuoka et al., 2016*), abbreviated *Tg(hsp70l:sflt1)*; *Tg(hsp70l:sflt4, cryaa-cerulean)bns82* (*Matsuoka et al., 2016*), abbreviated *Tg(hsp70l:sflt4)*; *Tg(−4.9sox10:EGFP)ba2* (*Carney et al., 2006*), abbreviated *Tg(sox10:EGFP)*; *Tg(sox10:mRFP)vu234* (*Kucenas et al., 2008*); *Tg(sox10:Gal4-VP16)sq9* (*Lee et al., 2013*); *Tg(ntla:Gal4-VP16)sq12* (*Lee et al., 2013*); *TgBAC(vegfab:gal4ff)bns273* (*Mullapudi et al., 2019*); *Tg(glut1b:mCherry)sj1* (*Umans et al., 2017*); *Tg(plvapb:EGFP)sj3* (*Umans et al., 2017*), abbreviated *Tg(plvap:EGFP)*; *Et(cp:EGFP)sj2* (*Henson et al., 2014*); *Tg(mpeg1.1:Dendra2)uwm12* (*Harvie et al., 2013*); *Tg(lyz:EGFP)nz117* (*Hall et al., 2007*); *Tg(lyve1:DsRed)nz101* (*Okuda et al., 2012*); *vegfaabns1* (*Rossi et al., 2016*); *vegfabbns92* (*Rossi et al., 2016*); *vegfchu6410* (*Helker et al., 2013*); *vegfcum18* (*Villefranc et al., 2013*); and *vegfdbns257* (*Gancz et al., 2019*). Adult fish were maintained on a standard 14 hr light/10 hr dark daily cycle. Fish embryos/larvae were raised at 28.5℃. To prevent skin pigmentation, 0.003% phenyl-thiourea (PTU) was used beginning at 10–12 hpf for imaging. Fish larvae analyzed at 10 dpf were transferred to a tank containing approximately 250 mL water supplemented with 0.003% PTU (up to 25 larvae/tank) and fed with Larval AP100 (<50 microns dry diet, Zeigler) starting at 5 dpf.

## Generation of new transgenic lines

The *TgBAC(vegfc:gal4ff)bns270*, *TgBAC(vegfd:gal4ff)lri95*, and *TgBAC(vegfaa:gal4ff)lri96* fish lines were generated by following the standard BAC recombineering protocol described previously (*Bussmann and Schulte-Merker, 2011*). Briefly, BAC clones CH211-83E15, CH211-79O8, and CH211-37K5 (BACPAC Resource Center) were engineered to replace the *vegfc*, *vegfd*, and *vegfaa* start codon with *gal4ff*, respectively. The engineered constructs were injected into one-cell stage eggs together with transposase mRNA transcribed in vitro.

## Genotyping of mutants

Genotyping of *vegfdbns257* mutant fish was performed as previously described (*Gancz et al., 2019*). Genotyping of *vegfaabns1* and *vegfabbns92* mutant fish was performed by high-resolution melt analysis of PCR products using the following primers:

vegfaa bns1 forward: 5'- CGAGAGCTGCTGGTAGACATC −3'
vegfaa bns1 reverse: 5'- GGATGTACGTGTGCTCGATCT −3'
vegfab bns92 forward: 5'- GTGCTGGGTGCTGCAATG −3'

vegfab bns92 reverse: 5'- CCAAGGTAATGTTGTATGTGACG −3'

Genotyping of *vegfc*[hu6410] and *vegfc*[um18] mutant fish was performed by standard PCR using the following primers:

vegfc hu6410 WT allele forward: 5'- CTTTCATCAATCTTGAACTTTT −3'
vegfc hu6410 WT allele reverse: 5'- AAACTCTTTCCCCACATCTA −3'
vegfc hu6410 mutant allele forward: 5'- GATGAACTCATGAGGATAGTTT −3'
vegfc hu6410 mutant allele reverse: 5'- TAAATTAATAGTCACTCACTTTACT −3'
vegfc um18 WT allele-specific forward: 5'- CTTCCTGCAGCTGTTTGTCAATAC −3'
vegfc um18 mutant allele-specific forward: 5'- CTTCCTGCAGCTGTTTGTCAATAT −3'
vegfc um18 common reverse: 5'- GCAACAATAAGGTGTTCCATTCGTG −3'.

## High-resolution melt analysis

A CFX96 Touch Real-Time PCR Detection System (Bio-Rad) was used for the PCR reactions and high-resolution melt analysis. Precision Melt Supermix for high-resolution melt analysis (Bio-Rad) was used in these experiments. PCR reaction protocols were: 95℃ for 2 min, 46 cycles of 95℃ for 10 s, and 60℃ for 30 s. Following the PCR, a high-resolution melt curve was generated by collecting Eva-Green fluorescence data in the 65–95℃ range. The analyses were performed on normalized derivative plots.

## Metronidazole (MTZ) treatment

For the cell ablation experiments in this study, MTZ substrate (Sigma, M3761) was used at 10 mM concentration dissolved in egg water containing 1% DMSO. Prior to treatment with 10 mM MTZ or 1% DMSO, embryos were manually dechorionated with forceps and incubated with freshly prepared 10 mM MTZ or 1% DMSO in egg water. Embryos were treated with 10 mM MTZ or 1% DMSO for indicated periods of time until imaging analyses without replacing solutions.

## Heat-shock treatments

Fish embryos/larvae raised at 28.5℃ were subjected to a heat shock at 37℃ for 1 hr by replacing egg water with pre-warmed (37℃) egg water. After each heat shock, the fish embryos/larvae were kept at room temperature for 10 min to cool down and then incubated at 28.5℃. Heat-shock experiments were performed as follows: Tg(kdrl:EGFP)[s843], Tg(kdrl:EGFP)[s843];Tg(hsp70l:sflt1, cryaa-cerulean)[bns80], and Tg(kdrl:EGFP)[s843];Tg(hsp70l:sflt4, cryaa-cerulean)[bns82] animals were subject to multiple heat shocks every 8 hr from 41 to 72 hpf.

## Immunohistochemistry

Immunohistochemistry was performed by following standard immunostaining procedures. Fish embryos/larvae were fixed in pH adjusted (pH 7.0) 4% paraformaldehyde (PFA)/phosphate buffered saline (PBS) overnight at 4℃ and dehydrated through immersion in methanol serial dilutions (50%, 75%, then 100% methanol three times, 10 min each) at room temperature (RT). The dehydrated samples were stored in 100% methanol at −20℃ at least for 2 hr before use. The samples were rehydrated through immersion in methanol serial dilutions (75%, 50% then 25% methanol, 10 min each) at RT and washed in 1% PBST (1% Triton X-100 in 0.1M PBS) followed by permeabilization in 10 µg/ml Proteinase K in 1% PBST at RT for 15 min. Samples were blocked at RT for 2–4 hr in the blocking solution containing 0.5% bovine serum albumin in 1% PBST and incubated with primary antibodies diluted in the blocking solution at 4℃ overnight. The following primary antibodies were used: chicken anti-GFP (Aves Labs at 1:1000), rabbit anti-DsRed (Clontech Labs at 1:300), and mouse anti-Claudin 5 (4C3C2, Thermo Fisher Scientific at 1:500). The next day, the samples were washed six times in 1% PBST at RT for 10 min each and incubated with Alexa Fluor dyes conjugated secondary antibodies (Invitrogen at 1:500) diluted in the blocking solution at 4℃ overnight. The samples were washed six times in 1% PBST at RT for 10 min each and then imaged.

## In situ hybridization

Whole-mount in situ hybridization was performed as previously described (*Thisse and Thisse, 2008*). Digoxigenin (DIG)-labeled cRNA antisense probes for *vegfaa*, *vegfab*, *vegfc*, and *vegfd* were

transcribed in vitro using a DIG RNA labeling kit (SP6/T7, Sigma). Targeted DNA sequences of each gene were amplified using the following gene-specific primers containing the T7 (5'- taatacgactcac-tatagg −3') or SP6 (5'- atttaggtgacactata −3') promoter sequence (underlined):

Vegfaa forward: 5'- GGCAGAACTCTGCATGAAGGAAGG −3'
Vegfaa reverse: 5'- gtaatacgactcactatagggCGTCAGTATAATACACTTTACCAC −3'
Vegfab forward: 5'- ATGTCTAACTTGCTTTCTGAGACC −3'
Vegfab reverse: 5'- gatttaggtgacactatagTCACCTCCTTGGTTTGTCACATCTG −3'
Vegfc forward: 5'- ATGCACTTATTTGGATTTTCTGTC −3'
Vegfc reverse: 5'- gatttaggtgacactatagTTAGTCCAGTCTTCCCCAGTATGTG −3'
Vegfd forward: 5'- ATGAAGAAACAGAAATGTGCTGGAC −3'
Vegfd reverse: 5'- gatttaggtgacactatagTCACGTATAGTGTAGTCTGTGTGTC −3'

## Plasmid and crRNA injections

For *ntla:EGFP* plasmid injection, the plasmid was generated by cloning the EGFP coding sequence downstream of approximately 2.1 kb *ntla* regulatory elements (*Harvey et al., 2010*; *Lee et al., 2013*) flanked by Tol2 sites. The 2.1 kb *ntla* regulatory elements were amplified by PCR from genomic DNA of zebrafish embryos using the following primers: *ntla* promoter forward: 5'- attaatcgatA TACAATTCCTTTGTGCTGTTGCAACAC −3' *ntla* promoter reverse: 5'- attagaattcATTTCCGA TCAAATAAAGCTTGAGAT −3'.

*Cla I* and *EcoR I* restriction enzyme sites (underlined) were incorporated into the primers to allow cloning. The engineered construct (12 pg per embryo) was injected together with transposase mRNA (*tol2* mRNA, 25 pg per embryo) into Tg(UAS-E1b:NfsB-mCherry)[c264] one-cell stage embryos that carry the Tg(ntla:Gal4-VP16)[sq12], TgBAC(vegfab:gal4ff)[bns273], TgBAC(vegfc:gal4ff)[bns270], or TgBAC(vegfd:gal4ff)[lri95] transgene. The injected embryos were fixed in 4% PFA/PBS (pH 7.0) overnight at 4°C and subjected to immunohistochemistry by following the immunostaining procedures as described without dehydrating fixed embryos through immersion in methanol serial dilutions. Anti-GFP and anti-DsRed antibodies were used to immunostain these embryo samples.

For *ccbe1* crRNA injection, crRNA:tracrRNA:Cas9 ribonucleoprotein (RNP) complexes were prepared as previously described (*Hoshijima et al., 2019*). Briefly, duplex buffer (Integrated DNA Technologies, IDT) was used to make 100 μM stock solution of individual crRNA and tracrRNA. Each of crRNA stock solution (100 μM) was mixed with an equal volume of tracrRNA (100 μM) and annealed in a PCR machine as previously described (*Hoshijima et al., 2019*) to generate crRNA:tracrRNA duplex complexes (50 μM). 50 μM crRNA:tracrRNA duplex solution was diluted in duplex buffer to generate 25 μM stock solution, which was stored at −20°C. The following sequences of *ccbe1* crRNAs were used: *ccbe1* crRNA1: TTCTCCTCTCGGAAAGTCCA; *ccbe1* crRNA2: TACCCGTGCG TAAAGTCCAC; *ccbe1* crRNA3: CTTCTGGAATACACTGACCC.

25 μM stock solution of Cas9 protein (S.p. Cas9 nuclease, V3, IDT) was prepared as previously described (*Hoshijima et al., 2019*). Injection cocktails (4 μL) that contained the crRNA:tracrRNA: Cas9 RNP complexes were prepared as follows: 0.4 μL 25 μM crRNA1:tracrRNA, 0.4 μL 25 μM crRNA2:tracrRNA, 0.4 μL 25 μM crRNA3:tracrRNA, 1.2 μL 25 μM Cas9 protein stock, 1.2 μL H$_2$O, and 0.4 μL 0.5% phenol red solution (Sigma). Control injection cocktails (4 μL) without cRNAs were prepared as follows: 1.2 μL 25 μM Cas9 protein stock, 2.4 μL H$_2$O, and 0.4 μL 0.5% phenol red solution. Both injection cocktails were freshly prepared on the day of injection. Prior to microinjection, both injection cocktails were incubated at 37°C for 5 min and then kept at room temperature. Approximately 2 nL of injection cocktails was injected into the cytoplasm of one-cell stage embryos.

## Confocal and stereo microscopy

A Leica TCS SP8 confocal laser scanning microscope (Leica) was used for live, immunofluorescence, and transmitted differential interference contrast imaging. Fish embryos and larvae were anaesthetized with a low dose of tricaine, embedded in a layer of 1% low-melt agarose in a glass-bottom Petri dish (MatTek), and imaged using a 10X dry or 25X water immersion objective lens. For microangiography, Qdot 655 streptavidin-conjugated nanocrystals (Thermo Fisher Scientific) were injected into the common cardinal vein and imaged after 30 min. An SMZ-18 stereomicroscope (Nikon) was used for brightfield images of anaesthetized fish.

## Transmission electron microscopy

10 dpf wild-type larvae were anesthetized with tricaine and fixed by immersion in 2.5% glutaraldehyde/4% PFA/0.2 M sodium cacodylate (pH 7.4) for 2 days at 4°C. Samples were post-fixed in 1% osmium tetroxide for 1 hr and incubated in 1% uranyl acetate in Maleate buffer for 1 hr. Samples were dehydrated through immersion in methanol serial dilutions and embedded in epoxy resin. Ultrathin sections of 85 nm were cut with a diamond knife, collected on copper grids, and stained with uranyl acetate and lead citrate. Images were captured using a transmission electron microscope (FEI Tecnai G2 Spirit BioTWIN).

## Quantification of DLV, PCeV, MCeV, and MsV formation

Fish carrying the *Tg(kdrl:EGFP)*[s843] reporter were used for this quantification. Fish embryos/larvae at all indicated developmental stages were analyzed for the presence or absence of the DLV. DLV lengths were measured in 2 and 3 dpf DLV-forming embryos/larvae by defining the region of interest (ROI) using the polyline tool in Leica Application Suite X (LAS X) software. To quantify and compare bilaterally formed PCeV, MCeV, and MsV, the following criteria were used to score the extent of each vessel's formation: (1) Score 2 - when the bilateral vessels are both fully formed; (2) Score 1.5 - when the vessel on one side is fully formed, but that on the other side is partially formed; (3) Score 1 - when the vessel on one side is fully formed, but that on the other side is absent; (4) Score 0.5 - when the vessel on one side is partially formed, but that on the other side is absent; and (5) Score 0 - when the bilateral vessels are both absent.

## Analysis of Claudin-5 protein expression in the mCP vasculature

Claudin-5 is strongly expressed in non-fenestrated brain vascular endothelium, while only weakly, or nearly undetectable, expression was observed in fenestrated mCP vascular endothelium (*van Leeuwen et al., 2018*). To analyze Claudin-5 protein expression levels in fenestrated mCP vECs in WT, *vegfab*[bns92], *vegfc*[hu6410], *vegfd*[bns257] mutants alone, and in combinations, 10 dpf mutant larvae carrying the *Tg(kdrl:EGFP)*[s843] reporter were immunostained for GFP and Claudin-5. Images were taken under the same microscopy settings across genotypes for comparison. The Claudin-5 immunostaining signals in the DLV and PCeV were qualitatively analyzed after imaging.

## Analysis of *Tg(glut1b*:mCherry) and *Tg(plvap*:EGFP) expression in the mCP vasculature

The *Tg(glut1b:mCherry)* reporter gene is strongly expressed in the vECs that comprise non-fenestrated brain blood vessels, while only weakly, or nearly undetectable, expression was observed in fenestrated CP vascular endothelium (*Umans et al., 2017*). In contrast, the *Tg(plvap:EGFP)* reporter gene is strongly expressed in fenestrated CP vascular endothelium (*Umans et al., 2017*). To analyze *Tg(glut1b:mCherry)* and *Tg(plvap:EGFP)* reporter expression patterns in the cranial vasculature of *vegfab*[bns92], *vegfc*[hu6410], and *vegfd*[bns257] mutants, each heterozygous mutant fish carrying both of the *Tg(glut1b:mCherry)*[sj1] and Tg(*plvapb:EGFP*)[sj3] transgenes were outcrossed with the corresponding heterozygous counterparts. The double transgenic progeny carrying *Tg(glut1b:mCherry)*[sj1];Tg(*plvapb:EGFP*)[sj3] reporters were fixed at 10 dpf and immunostained with anti-GFP and anti-DsRed antibodies. Images were taken under the same microscopy settings across genotypes for comparison. The *Tg(glut1b:*mCherry) and *Tg(plvap:*EGFP) reporter expression in the DLV and PCeV was qualitatively analyzed after imaging.

## Quantification of intersegmental vessel (ISV) formation

*Tg(kdrl:EGFP)*[s843] reporter fish were used to quantify ISV formation in WT, single mutants of *vegfab*[bns92], *vegfc*[hu6410], and *vegfd*[bns257], and double mutant animals of *vegfab*[bns92];*vegfc*[hu6410] and *vegfab*[bns92];*vegfd*[bns257] at 96 hpf. For each fish, 10 ISVs in the trunk region of 2 somites directly anterior, and two somites directly posterior, to the anal opening were analyzed. The number of ISVs, including truncated ones, was quantified.

## Co-localization analysis of *Tg(sox10:*EGFP)-positive or *Tg(mpeg1.1:* Dendra2)-positive meningeal cells

To analyze *TgBAC(vegfab:*gal4ff)$^{bns273}$-, *TgBAC(vegfc:*gal4ff)$^{bns270}$- or *TgBAC(vegfd:*gal4ff)$^{lri95}$-positive meningeal cells, *TgBAC(vegfab:gal4ff)$^{bns273}$*, *TgBAC(vegfc:gal4ff)$^{bns270}$* and *TgBAC(vegfd: gal4ff)$^{lri95}$* fish carrying *Tg(UAS-E1b:NfsB-mCherry)$^{c264}$* and *Tg(−4.9sox10:EGFP)$^{ba2}$* or *Tg(UAS-E1b: NfsB-mCherry)$^{c264}$* and *Tg(mpeg1.1:Dendra2)$^{uwm12}$* transgenes were imaged at 48 hpf. High-magnification z-plane confocal images were collected at every 1 μm interval using a 25X water immersion objective lens to visualize NTR-mCherry$^{+}$ cells present in the dorsal surfaces of the brain meninges, particularly near the DMJ. These images were analyzed for the co-localization of NTR-mCherry$^{+}$ cells and *Tg(sox10:*EGFP)$^{+}$ neural crest-derived meningeal cells or *Tg(mpeg1.1:*Dendra2)$^{+}$ macrophages at every single z-plane.

## Analysis of FGP development in the dorsal surfaces of the brain meninges

*Tg(kdrl:EGFP)$^{s843}$;Tg(lyve1:DsRed)$^{nz101}$* double Tg reporter fish were used to visualize blood vessels and FGPs simultaneously in the dorsal surfaces of the brain meninges. To determine the developmental relationships between FGPs and mCP vasculature, *vegfab$^{+/-}$*, as well as *vegfc$^{+/-}$; vegfd$^{-/-}$*, adult fish that carried these Tg reporters were incrossed. The double transgenic progeny from these incrosses was imaged at 120 hpf and genotyped afterwards. For validation of *ccbe1* crRNA injection experiments, *Tg(kdrl:EGFP);Tg(lyve1:DsRed)* double Tg fish were outcrossed with wild-type zebrafish AB strains. The progeny were injected at the one-cell stage, and the double transgenic progeny were imaged at 120 hpf. For experiments involving combined gene inactivation of *ccbe1* and *vegfab*, *vegfab$^{+/-}$* adult fish that carried the *Tg(kdrl:EGFP);Tg(lyve1:DsRed)* reporters were incrossed, and progeny were injected at the one-cell stage. The double transgenic progeny were imaged at 96 hpf and genotyped afterwards. For all the experiments, a *Tg(lyve1:*DsRed)$^{+}$ bilateral loop of FGPs that formed in the dorsal meningeal surfaces of the optic tectum were analyzed for presence or complete absence.

## Statistical analysis

Statistical differences for mean values among multiple groups were determined using a one-way analysis of variance (ANOVA) followed by Tukey's or Dunnett's multiple comparison test. Fisher's exact test was used to determine significance when comparing the degree of penetrance of observed phenotypes. The criterion for statistical significance was set at $P<0.05$. Error bars are SD.

## Acknowledgements

We thank Dr. Didier Stainier for his generous support during the project; Drs. Nathan Lawson, Michael Taylor, and Saulius Sumanas for kindly providing us with fish lines; and Drs. Judith Drazba, Gauravi Deshpande, and Mei Yin for imaging. This work was supported by funding from the National Institutes of Health (R01 NS117510) and start-up funds from the Cleveland Clinic Foundation to RLM.

## Additional information

### Funding

| Funder | Grant reference number | Author |
|---|---|---|
| National Institute of Neurological Disorders and Stroke | R01 NS117510 | Ryota L Matsuoka |
| Lerner Research Institute, Cleveland Clinic | | Ryota L Matsuoka |

The funders had no role in study design, data collection and interpretation, or the decision to submit the work for publication.

## Author contributions
Sweta Parab, Resources, Data curation, Software, Formal analysis, Validation, Investigation, Visualization, Methodology, Writing - original draft, Writing - review and editing; Rachael E Quick, Resources, Data curation, Software, Validation, Visualization, Methodology, Writing - review and editing; Ryota L Matsuoka, Conceptualization, Resources, Data curation, Software, Formal analysis, Supervision, Funding acquisition, Validation, Investigation, Visualization, Methodology, Writing - original draft, Project administration, Writing - review and editing

## Author ORCIDs
Sweta Parab http://orcid.org/0000-0002-9932-5117
Rachael E Quick http://orcid.org/0000-0003-2301-7238
Ryota L Matsuoka https://orcid.org/0000-0001-6214-2889

## Ethics
Animal experimentation: This study was performed in strict accordance with the recommendations in the Guide for the Care and Use of Laboratory Animals of the National Institutes of Health. All zebrafish work was approved by the Cleveland Clinic's Institutional Animal Care and Use Committee under the protocol number 2018-1970. Every effort was made to minimize suffering and distress of the animals used throughout this study.

## Decision letter and Author response
Decision letter https://doi.org/10.7554/eLife.64295.sa1
Author response https://doi.org/10.7554/eLife.64295.sa2

# Additional files
## Supplementary files
• Transparent reporting form

## Data availability
All data generated or analysed during this study are included in the manuscript and supporting files. Source data files have been provided for Figures 1 and 4.

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
