## [Decision Letter]

[Editors' note: this paper was reviewed by Review Commons.]

**Acceptance summary:**

Your study makes an important contribution to the understanding of blood vessel formation and in particular, the role this plays in the developing CNS.

---

## [Author Response]

We would like to thank all of the reviewers for their thoughtful, critical, and constructive feedback on our manuscript. These comments greatly helped us to improve our current study.

Reviewer #1:In this manuscript Parab et al. investigate the molecular mechanisms governing the formation of fenestrated blood vessels in the myelencephalic choroid plexus. The authors use state of the art zebrafish transgenic lines and genetic tools. In general the manuscript is well written, data is nicely presented and controls are properly used an shown. The study is interesting as it shows the heterogeneity in the regulation of blood vessel formation. Based on the expression pattern of the different Vegf family ligands, this study also indicates the effect of local cues on blood vessel formation. However, mechanistics insight of how fenestration of those vessels is regulated are missing.

We appreciate this reviewer for her/his constructive comments and suggestions to improve this study. We address each of the points raised by the reviewer below.

Major comments:The analysis of markers for fenestrated vessels is not clear. The authors show that vegfab^-/-^, vegfab-/vegfc^-/-^ and vegfab^-/-^vegfd^-/-^ show "no DLV" phenotypes. However, in Figure S3C they then go and study Claudin5 expression in DLV of those mutants. If at 10dpf around 30% of vegfab^-/-^, 65-70% of the vegfab^-/-^vegfc^-/-^, and aprox 80% of the vegfab^-/-^vegfd^-/-^, do not have DLV, what ZF mutants do the authors use to analyze claudin5 expression in the DLV? do they analyze the remaining percentage of the ones where the DLV is formed? and if so, what is the significance of this analysis? Could it be that in those mutants there are compensatory mechanisms occurring that assure DLV formation and expression of BBB markers? Could the authors clarify this? if so, a more appropriate analysis would be to analyze BBB markers in mutants with partial DLV.

We appreciate this comment and agree with the reviewer that our Claudin-5 expression analyses of only the mutants that formed the DLV could misrepresent the results due to the compensatory mechanisms among the Vegf ligands and possibly other unidentified factors. To address this concern, we have now clarified in both the Results and Materials and methods sections of the revised manuscript that this analysis was feasible only for the mutants that formed the DLV. Secondly, we performed additional experiments to examine additional fenestration and BBB endothelial markers by crossing each of the *vegf* mutants with the *Tg(plvap:EGFP);Tg(glut1b:mCherry)* double transgenic line. We have now added these completed analyses of the additional endothelial markers conducted in 10 dpf WT, *vegfab^-/-^*, *vegfc^-/-^*, and *vegfd^-/-^* larvae to the revised manuscript (Figure 5). Lastly, we have described the limitations of these experimental analyses in the Discussion section by incorporating the possibility of compensatory actions that were brought to our attention by the reviewer.

While the rest of the conclusions are convincing, it still remains to be understood what regulates fenestration of those vessels compared to non-fenestrated ones. It would be interesting to further investigate and this still unknown question.

We fully agree with the reviewer that further investigation into regulators of mCP vascular fenestration will be a crucial next step and remains an important question to further address. Our perspectives with regards to this question are described in greater detail below in the response to this reviewer’s last comment.

Reviewer #1 (Significance (Required)):Nature and significance of the advance: the study shows significant advance in understanding the molecular mechanisms of brain vascularization

We appreciate this positive comment highlighting the significance of our study in the field.

Work in context of existing literature: based on the previous published literature, this study provides mechanistic insights into regulation of blood vessel formation in the choroid plexus. It would be more relevant if the authors would provide more data on why those vessels are fenestrated and the other not. What controls this fenestration?

To gain more insight into a potential role for Vegfs in the development of vessel fenestration in the mCP vasculature, we conducted expression analyses of additional fenestration and BBB endothelial markers in 10 dpf WT, *vegfab^-/-^*, *vegfc^-/-^*, and *vegfd^-/-^* larvae as described above. However, no obvious differences were observed between WT and each of these mutants in the expression patterns of the *Tg(plvap:EGFP);Tg(glut1b:mCherry)* reporters which we utilized for this analysis. These observations combined with our analysis of Claudin-5 protein expression in these single and double mutants indicate that the lack of these Vegf ligands does not lead to an absence, or a conversion, of the mCP endothelial fenestrated state, although our analyses were conducted only for the mutants that formed the DLV and PCeV.

Considering these observations and the recent reports in the field, we have now formulated several hypotheses aimed at uncovering the molecular mechanisms underlying the fenestration of mCP vasculature. In light of recent studies (Wang et al., 2019; Anbalagan et al., 2018), we speculate that vessel fenestration in the CP may require combined actions of 1) endogenous inhibitor(s) of canonical Wnt/b-catenin signaling, 2) vascular permeability/inflammatory protein(s), and/or 3) signaling molecules inhibiting endothelial tight junction formation. These combined actions allow vascular endothelial cells in the CP to 1) maintain low levels of Wnt/b-catenin signaling, 2) exhibit increased vascular permeability, and 3) form loose endothelial connections, as compared to those forming the BBB in the other areas of the brain. Our future research efforts will test these hypotheses in order to define molecular players involved in CP vessel fenestration in the most comprehensive manner possible.

Reviewer #2:Parab et al. use the zebrafish model to study the requirement of angiogenic factor VEGFs in choroid plexus (CP) vascularization. Using a series of elegant genetic experiments and anatomical imaging, the authors clearly demonstrate that the combinations of vegfab, vegfc, and vegfd deficiency cause defective CP vascularization, as well as dorsal longitudinal vein (DLV) and posterior cerebral veins (PCeV) which are functionally integrated in the CP vasculature.The question of what controls the CP vascularization with highly permeable fenestrate is clearly an important one, and this paper is impressive in terms of the thoroughness of genetic approaches brought to bear on it. The experiments are nicely carried out with quantitative measurements. However, it is not clear that the data presented are enough to explain the involvement of VEGFs in the CP vascularization. The important question is whether lack of the CP vasculature in the vegf mutants may be secondary to the DLV and PCeV phenotypes. It is not clear what directs the recruitment of these vessels into the CP.

We appreciate this reviewer for her/his positive and encouraging comments highlighting both the quality and significance of our study. We also appreciate the reviewer’s thoughtful comments on the requirements for Vegfs in CP vascularization. As the reviewer pointed out, there is a possibility that the DLV/PCeV development and mCP vascularization could have separate molecular requirements for angiogenesis and that the lack of mCP vascularization in the *vegf* mutants could be secondary to the DLV/PCeV defects. Although only small segments of the DLV and PCeV are present in close proximity to the mCP, these two vessels exhibit molecular signatures of fenestrated endothelial cells along the entire vessel – even in the brain/meningeal regions outside of the mCP. In mice, fenestrated endothelial cells are observed in the CPs, but do not appear to be present in their directly adjacent tissues (Wang et al., 2019), although it is possible that a blood vessel(s) giving rise to these CP fenestrated vasculature is present on the meningeal compartments and displays molecular signatures of fenestrated endothelial cells as observed similarly in zebrafish.

As previously reported (Bill B.R. et al., PLOS ONE 3:e3114, 2008), the DLV is clearly the only blood vessel that supplies blood to the mCP (at least up to 10 dpf) based on the blood flow observed under transmitted differential interference contrast (DIC) imaging microscopy, and this blood flow leaves the mCP through bilateral PCeVs. These closely connected endothelial cells comprising the functionally integrated mCP vasculature acquire similar endothelial features such as fenestration for the purpose of achieving a shared goal as the functional mCP vasculature in zebrafish. As the reviewer nicely summarized, our findings demonstrate that specific combinations of Vegfab, Vegfc, and Vegfd deficiency result in defective mCP vascularization and the lack of the DLV and PCeV. The DLV and PCeV converge to form the mCP vasculature, and all of these vessels exhibit the shared fenestrated endothelial state. Thus, we expect that the mCP vasculature is an integral constituent of the connected DLV and PCeV. Further dissections of these Vegf ligands’ functions in mCP vascularization versus DLV/PCeV formation would require more sophisticated strategies, experimental tools and approaches, even if feasible. However, since the reviewer’s point leaves an important question open and suggests another possible interpretation of the results, we have now added the relevant discussions to the revised manuscript.

With regard to the question of what directs the recruitment of these vessels into the CP, we speculate that *vegfab* expressed in the mCP epithelial cells plays an instructive role. However, there are currently no CP-specific promoter or zebrafish tools available in the community that allow us to perform genetic manipulation in the CP specifically. The most specific and powerful tool available in the community is the enhancer trapped *Et(cp:EGFP)^sj2^* line. Although we have been trying to convert this pre-existing EGFP enhancer trap line into a Gal4 (KalTA4) line by following the recently published method (Auer T.O. et al., Genome Res. 24:142-153, 2014; Auer T.O. et al., Nat Protoc 9:2823-2840, 2014), the establishment of this new line would require substantial time, even if successful.

Alternatively, we attempted two-photon laser-mediated cell ablation that specifically targeted the *vegfab*-expressing mCP epithelial cells directed by the *TgBAC(vegfab:gal4ff);Tg(UAS:EGFP-CAAX)* line in order to investigate a role for these cells in DLV and PCeV formation. However, cell ablation of these cells prior to DLV sprouting (~42 hpf) was limited by the fact that only the visible EGFP-expressing cells can be targeted at that stage and that *vegfab*-expressing mCP epithelial cells continue to emerge while the DLV is extending toward the mCP. Given the concurrent and rapid development of the DLV and mCP, a two-photon laser-mediated cell ablation approach does not seem to be the best option.

Considering these current technical limitations, we now believe that establishing a CP-specific zebrafish tool by converting the existing *Et(cp:EGFP)^sj2^* line into a Gal4 line will overcome some of these technical limitations and thus be an important next step. Successful generation of this new Gal4 line will allow ones to utilize the Gal4/UAS system to overexpress genes of interest in any form (eg. full-length, dominant-negative, constitutive-active, or truncated forms of genes) specifically in CP epithelial cells, thereby allowing for CP tissue-specific gene inactivation, overexpression, and rescue of mutants. Thus, the establishment of this sort of new CP-specific transgenic Gal4 line will be needed for us to further delve into the mechanisms of mCP vascular development.

1) Based on the observation that vegf mutants with DLV retain high expression of the permeability marker PLVP and low expression of BBB markers such as GLUT1 and Claudin-5, how the authors conclude that VEGFs do not influence DLV fenestration remains unclear. The authors cannot rule out the possibility that due to an incomplete penetrance these mutants may not have both DLV formation and fenestration phenotypes. The authors need to carry out in situ hybridization to examine the expression of vegfs in the vegf mutants with DLV.

We appreciate this comment and agree with the reviewer that our analyses of only the mutants that formed the DLV can misrepresent the results due to the incomplete penetrance and compensatory mechanisms that can occur in the *vegf* mutants we examined. Related to reviewer #1’s first major comments above, we have now clarified that this analysis was feasible only for the mutants that formed the DLV in both the Results and Materials and methods sections of the revised manuscript. We have also described the technical limitations and possible interpretations of these experimental analyses in the Discussion section by including the possibility of compensatory actions that were brought to our attention by this reviewer and reviewer #1.

With regards to *vegf* expression analyses in the *vegf* mutants that form the DLV, we conducted this analysis using our *vegf* BAC lines instead of performing *in situ* hybridization. We expect that this approach provides much more sensitive and higher resolution analyses than using *in situ* hybridization, which generally provides lower contrast and resolution due to higher background staining in the head of embryos/larvae at the relevant developmental stages of this analysis (eg. 48-72 hpf). We analyzed *vegfc* expression in *vegfab* mutants by examining the *TgBAC(vegfc:*gal4ff*);Tg(UAS:*EGFP-CAAX) reporter expression, which did not show any obvious differences in expression patterns and/or fluorescence signal intensity when compared to *vegfab* WT and Het sibling controls. Similarly, we observed no obvious difference in *TgBAC(vegfd:*gal4ff*);Tg(UAS:*EGFP-CAAX) reporter expression in *vegfab* mutants compared to their sibling controls. Lastly, our analysis of *TgBAC(vegfab:*gal4ff*);Tg(UAS:*EGFP-CAAX) reporter expression in *vegfc* mutants and their control siblings did not reveal any obvious difference between the mutants and controls. We have now added these results to the revised manuscript (Figure 6—figure supplement 2).

2) In Figure 1I and 1J, the authors need to clarify what expresses Claudin-5 in the absence of DLV in vegfab mutants. Is it possible that the vegfab mutants may fail to maintain VEGFR2 expression (kdrl:EGFP) but the mutants may still have DLV? It would be helpful if the authors could examine functional and morphological properties of DLV in the vegfab mutants using a fluorescent dye injection.

We appreciate this comment and have now clarified that there are Claudin-5-expressing cells which reside ventrally to the DLV and that they are visible in the presence or absence of the DLV. We also agree with the reviewer that examining whether a perfused, functional DLV is present or absent in *vegfab* mutants by fluorescent dye injections will clarify the concern regarding *Tg(kdrl:EGFP)* transgene downregulation/silencing in this mutant. To address this concern, we injected Qdot 655 streptavidin-conjugated nanocrystals through the common cardinal vein into 96 hpf WT and *vegfab^-/-^* larvae that carry the *Tg(kdrl:EGFP)* reporter. This injection experiment allowed us to visualize endothelial cell-specific *Tg(kdrl:*EGFP) reporter expression and perfused, functional blood vessels labeled by blood-circulating Qdot 655 nanocrystals. We observed that in all *vegfab^-/-^* mutants we examined (n=8) that lack the *Tg(kdrl:*EGFP)-positive DLV, Qdot 655 nanocrystals failed to label this vessel, illustrating the lack of a perfused, functional DLV. In addition, we examined blood flow and circulating blood cells in these WT and *vegfab^-/-^* larvae under transmitted differential interference contrast (DIC) imaging microscopy and observed no blood flow in the area where the DLV normally forms. We have now added these experimental results to the revised manuscript (Figure 1—figure supplement 2G–H”).

3) In Figure S2G-P, the authors need to clarify the descriptions of sflt1 and sflt4 mutant phenotypes. Do the sflt1 mutants have no significant penetrance (21%) of the "No DLV" phenotype but some mutants (79%) form relatively short DLV? Do the sflt4 mutants have a significant penetrance (33%) of the "No DLV" phenotype but some mutants (67%) form normal DLV?

We appreciate this suggestion. The reviewer’s interpretation is correct. We have now clarified the descriptions of these results in the revised manuscript as suggested by the reviewer.

4) In Figure 2G, the authors need to define "score."

We have now corrected this term into more defined titles in this and all the relevant graphs.

5) In Figure S5, the data are not enough to conclude that the mesoderm-derived meningeal fibroblasts express vegfs. The authors need to examine whether meningeal macrophages and/or glial cells express vegfs.

We appreciate this comment and agree with the reviewer that there are some other cell types that reside in meningeal compartments and that our current analysis cannot rule out the possibility that *vegfab*-, *vegfc*-, and/or *vegfd*-expressing cells are not mesoderm-derived meningeal fibroblasts. We have now examined the co-localization of meningeal macrophages and these *vegf*-expressing cells by crossing each of *TgBAC(vegfab:NTR-mCherry)*, *TgBAC(vegfc:NTR-mCherry)*, and *TgBAC(vegfd:NTR-mCherry)* fish with the *Tg(mpeg1.1:Dendra2)^uwm12^*transgenic reporter line that was previously shown to label macrophages in the brain and meninges (Ellett F. et al., *Blood* 117:e49-56, 2010). We have added these results to the revised manuscript (Figure 6—figure supplement 4).

In addition, to examine whether these *vegf*-expressing cells are mesoderm-derived meningeal fibroblasts, we injected a plasmid in which the promoter of the early mesoderm-specific gene, *ntla*, drives the expression of EGFP into 1-cell stage *TgBAC(vegfab:NTR-mCherry)*, *TgBAC(vegfc:NTR-mCherry)*, and *TgBAC(vegfd:NTR-mCherry)* embryos. To validate this approach, we injected this plasmid into 1-cell stage *Tg(ntla:gal4);Tg(UAS:NTR-mCherry)* embryos and observed that the EGFP and NTR-mCherry signals directed by the *ntla* promoter were mostly co-localized in the dorsal surfaces of the brain meninges. Using this *ntla:EGFP* plasmid injection approach, we found that NTR-mCherry^+^ meningeal cells marked by each of the *vegf* BAC lines were co-localized with EGFP^+^ mesoderm-derived meningeal cells. In addition, we took magnified confocal images of the meningeal cells labeled by *Tg(ntla:gal4)*, *TgBAC(vegfab:gal4ff)*, and *TgBAC(vegfd:gal4ff)* Gal4 drivers to show close similarity in their cell morphology. These results provide evidence that mesoderm-derived meningeal fibroblasts are a cell type that expresses these *vegf* genes. We have now added these results to the revised manuscript (Figure 6H– L” and Figure 6—figure supplement 5).

6) Combined with Figure S5 and Figure 5, these data suggest that ablation of vegfd- or vegfab-expressing cells leads to the failure of the DLV formation.

We appreciate this comment and agree with the reviewer that our data presented in the original manuscript are not sufficient to suggest that *vegfd*- or *vegfab*-expressing cells are mesoderm-derived meningeal fibroblasts and that these fibroblasts regulate DLV formation. Related to our responses to this reviewer’s comments above, we performed additional co-localization experiments as described and provide evidence that mesoderm-derived meningeal fibroblasts are a source of these Vegf ligands.

Reviewer #2 (Significance (Required)):As mentioned above, the authors' question of what controls the CP vascularization with highly permeable fenestrate is clearly an important one. The author's genetic approaches and anatomical imaging are significant.

We appreciate this reviewer for highlighting the significance of our study in the field.

Reviewer #3 (Evidence, reproducibility and clarity (Required)):I think this is a beautifully illustrated and solid piece of work and the combinatorial logic of the vegfs is shown meticulously, using several methods and numerous genetic models. In particular, I liked the validation of the autocrine function of vegfc, whereas I am not so familiar with potential pitfalls with the metronidazole-based deletion of putative producer cells. The authors have really achieved a very complete analysis of the redundancy of vegfs in mCP formation. I agree with most of their conclusions and the manuscript has excellent clarity.

We appreciate this reviewer for her/his favorable and encouraging comments highlighting both the quality and impact of our study.

I recommend publication after the issues below have been addressed.A major concern is the limited focus of the manuscript, and whether the blood vasculature outside of the area of interest is normal or if its defects are behind the findings. As mainly meningeal fibroblasts produce the vegf ligands, the authors should include a paragraph describing the lymphatic vascular phenotypes as these can affect the blood vascular defects. And what happens to the FGP/Mato cells after the gene deletions.

We appreciate this suggestion and agree with the reviewer that examining FGP/Mato cells (also called as muLECs or BLECs) in these *vegf* mutants will provide insight into the correlation between the mCP vascular defects and the formation of FGP/Mato cells. We have now examined the formation of FGP/Mato cells in the dorsal surfaces of the brain meninges using the *Tg(lyve1:DsRed)^nz101^* transgenic reporter line. The bottom line is that we did not find any correlation between mCP vascular defects and FGP/Mato formation defects since all of *vegfc;vegfd* double mutants (n=9), which completely lacked the FGP/Mato cells, formed the mCP vasculature, while all *vegfab* mutants that lacked the DLV (n=7) formed FGP/Mato cells similarly to their sibling WT controls (n=13). Intracranial lymphatic formation in zebrafish appears to begin at much later stages of development (9-10 dpf) (Castranova D. et al., Circ Res, doi: 10.1161/CIRCRESAHA.120.317372), thus mCP vascularization processes should not be affected by these intracranial lymphatics.

We have now added these experimental results and relevant discussions to our revised manuscript (Figure 3—figure supplement 1).

Furthermore, I am not entirely happy about the equation of Claudin-5 lacking vessels as fenestrated (also, the epithelium is highly Claudin-5 ositive, thus it is more difficult to be specific). There is no analysis of the fenestrae as such. The authors should use more than one marker, for example in the chapter on "Analysis of BBB state…". To my knowledge, plvap would be a good marker for the fenestrae.

We appreciate this suggestion and fully agree with the reviewer that further analyses using additional endothelial cell markers for the fenestrated and BBB state will improve this part of the manuscript. Related to reviewer #1’s comments, we analyzed *vegfab*, *vegfc*, and *vegfd* mutants using the *Tg(plvap:EGFP);Tg(glut1b:mCherry)* double transgenic reporter line and have now added these results to our revised manuscript (Figure 5).

Vegfc and vegfd are made as precursor proteins that require proteolytic activation, which in turn regulates receptor binding specificity. Do the authors know if for example vegfd needs to be processed to ameliorate the vegfa deletion phenotype?

We appreciate this comment and agree with the reviewer that understanding whether or not proteolytic activation of Vegfc and Vegfd contributes to the regulation of mCP vascularization, and how this is achieved at the cellular and molecular levels, will be an important and interesting question to address next. To gain insights into this process, we focused on investigating the requirements for the proteolytic processing of Vegfc since its proteolytic enzymes have been better characterized than those for Vegfd. Prior studies suggest that Ccbe1, Adamts3, and Adamts14 serve to proteolytically cleave and convert the pro-active form of Vegfc into its mature form during lymphangiogenesis in zebrafish (Hogan et al., 2009; Le Guen et al., 2014; Wang et al., 2020). Among these three proteins, *ccbe1* appears to be highly expressed in the developing brain and meningeal compartments based on its BAC transgenic reporter expression at 48 hpf (Wang et al., 2020), while *adamts3* BAC transgenic reporter expression appear to be very low at the same stage. Since 48 hpf is a developmental time point when the DLV is extending toward the mCP, *ccbe1* could be a strong candidate that serves to activate Vegfc in this area of the brain based on these published data.

To investigate a role for Ccbe1 in mCP vascularization, we took the approach of injecting

ribonucleoprotein (RNP) complexes that were composed of *ccbe1*-specific CRISPR RNA (crRNA):transactivating crRNA (tracrRNA):Cas9 protein at the 1-cell stage. This approach using crRNA:tracrRNA:Cas9 protein RNP complexes was previously shown to be a highly efficient method to achieve bi-allelic inactivation of target genes, which allows for the generation of F0 embryos that recapitulate homozygous mutant phenotypes (Hoshijima K. et al., *Dev Cell* 51:645-657). We recently tested this method and observed that injection of three different crRNA RNP complexes targeted to a single gene works in a much more efficient manner to generate F0 embryos that lack target gene function and recapitulate homozygous mutant phenotypes. Thus, we co-injected three different *ccbe1* crRNA RNP complexes in order to investigate its role in mCP vascularization. We validated this approach by examining the formation of FGP/Mato cells in the dorsal surfaces of the brain meninges since *ccbe1* mutants exhibit severe defects in this process as shown in *vegfc* null mutants (van Lessen et al., 2017).

We hypothesized that if Vegfc requires Ccbe1-mediated proteolytic activation, these triple *ccbe1* crRNA injected embryos will lack Vegfc activation, thus recapitulating *vegfc* null mutant phenotypes. Since we have shown that 54% of the *vegfc* null mutants we analyzed at 54 hpf failed to form the DLV, we examined *ccbe1* crRNA-injected embryos at 54 hpf and observed that a significant fraction (20%) of the *ccbe1* crRNA-injected embryos, but not Cas9 protein-injected control embryos, lacked the DLV.

In addition, to examine whether combined genetic inactivation of *ccbe1* and *vegfab* enhances DLV and PCeV formation defects as observed in *vegfab;vegfc* double mutants, we injected *ccbe1* triple crRNA RNP complexes into 1-cell stage progeny derived from *vegfab* heterozygous incrosses and analyzed them at 96 hpf – the same developmental stage at which we analyzed progeny derived from *vegfab;vegfc* double heterozygous fish incrosses (Figure 2A–G). Intriguingly, similar to what we observed in *vegfab;vegfc* double mutants, we found that *vegfab^-/-^* larvae injected with the *ccbe1* RNPs exhibit a markedly increased penetrance of the “No DLV” phenotype (52%) as compared to their *vegfab^-/-^* siblings injected with the control solution (23%) or the *ccbe1* RNPs-injected *vegfab^+/+^*sibling larvae (3%). Strikingly enhanced defects were also observed in PCeV formation in the *ccbe1* RNPs-injected *vegfab^-/-^* larvae. These results indicate that Ccbe1 and Ccbe1-mediated proteolytic activation of Vegfc are critical for DLV and PCeV formation.

We have now added all of these results to our revised manuscript (Figure 4).

The discussion is quite short and draws a few general conclusions that are important for our understanding. However, it leaves open the question of why vagfaa is not important for the formation of a fenestrated mCP, if in adult zebrafish it is necessary for the maintenance of mCP fenestrae. This is an issue that should be at least discussed.

We appreciate this comment. There is currently no direct evidence of Vegfaa’s function in maintaining mCP fenestrae or in the formation of fenestrated mCP vasculature in zebrafish. However, we agree with the reviewer that our findings leave these questions open, which will be important to address next. Since *vegfaa* mutants are known to exhibit severe early vascular defects that prevent us from understanding its precise role in mCP vascularization, we generated a new BAC transgenic line, *TgBAC(vegfaa:gal4ff)*, and performed expression analyses and chemogenetic cell ablation using this line, as conducted for the other *vegf* BAC lines (Figures 6 and 5E–M). We found that ablation of Vegfaa-expressing cells does not lead to compromised formation of the DLV and PCeV. We have now added all of these results to our revised manuscript (Figure 8).

In addition, as suggested by the reviewer, we have now expanded our discussions in the revised manuscript by including the discussions of these and another open questions that we deem important and also those of new experimental results obtained during manuscript revision.

Minor:Please do not use abbreviations, such as CP or vEC in the Highlights, as they are not explained yet in this part.

We appreciate this suggestion and have removed the Highlights.

Reviewer #3 (Significance (Required)):This manuscript provides an important addition to our understanding of how the vegf family of growth factors works, and it is especially interesting for developmental and vascular biologists.

We appreciate the reviewer for highlighting the significance of this study in the fields of developmental and vascular biology.